# Combining Tree-Search, Generative Models, and Nash Bargaining Concepts in Game-Theoretic Reinforcement Learning

## Abstract

Algorithms that combine deep reinforcement learning and search to train agents, such as AlphaZero, have demonstrated remarkable success in producing human-level game-playing AIs for large adversarial domains. We propose a like combination that can be applied to general-sum, imperfect information games, by integrating a novel search procedure with a population-based deep RL training framework. The outer loop of our algorithm is implemented by Policy Space Response Oracles (PSRO), which generates a diverse population of rationalizable policies by interleaving game-theoretic analysis and deep RL. We train each policy using an Information-Set Monte-Carlo Tree Search (IS-MCTS) procedure, with concurrent learning of a deep generative model for handling imperfect information during search. We furthermore propose two new meta-strategy solvers for PSRO based on the Nash bargaining solution. Our approach thus combines planning, inferring environmental state, and predicting opponents' strategies during online decision-making. To demonstrate the efficacy of this training framework, we evaluate PSRO's ability to compute approximate Nash equilibria in benchmark games. We further explore its performance on two negotiation games: Colored Trails, and Deal-or-No-Deal. Employing our integrated search method, we conduct behavioral studies where human participants negotiate with our agents ($N = 346$). We find that search with generative modeling finds stronger policies during both training time and test time, enables online Bayesian co-player prediction, and can produce agents that achieve comparable social welfare negotiating with humans as humans trading among themselves.

## 1 Introduction

Computer game research has witnessed tremendous progress over the past decade, marked prominently by the development of human-level game-playing bots in the games of Go (Silver et al., 2018), Poker (Brown et al., 2020; Schmid et al., 2021b), and Diplomacy (Bakhtin et al., 2023). Two broad algorithmic techniques are primarily responsible for this success: (1) deep reinforcement learning (RL) and (2) game-tree search. Deep RL methods are capable of training quality value functions or policies represented by neural nets which generalize well across large state spaces. Search techniques such as Monte-Carlo Tree Search (MCTS) (Browne et al., 2012) leverage computational resources at decision time to improve the strength of a strategy. AlphaZero (Silver et al., 2018) provides an elegant framework that coherently combines the power of both methods: a deep policy-and-value network (PVN) is trained using self-play trajectories generated by MCTS, and the updated PVN further guides the search procedure and improves the quality of the trajectory data. By iteratively training the PVN and simulating self-play matches, AlphaZero produces progressively stronger play, which eventually surpass professional human players without any human data. Outside recreational game domains, AlphaZero-style methods also achieved remarkable successes in discovering faster matrix completion methods (Fawzi et al., 2022) and sorting algorithms (Mankowitz et al., 2023).

AlphaZero was originally designed to master large, adversarial, perfect-information games. There are several barriers to generalization of this approach to general-sum, imperfect information domains. First, self-play

training is specifically geared to two-player zero-sum domains and implicitly depends on transitivity of the game (Balduzzi et al., 2019). For games that are not purely adversarial, issues like *equilibrium selection* appear: agents trained entirely through self-play optimize to their opponents at training time, thus may not perform well to opponents at test time, which may correspond to alternative equilibria. In cooperative settings like coordination or common-interest games, this issue can be alleviated by publishing the algorithms and random seeds as mutual knowledge among players (Foerster et al., 2019; Lerer et al., 2020). However, this is generally an unreasonable assumption for games involving mixed cooperative and competitive elements. Population-based training methods provide one approach to dealing with this issue. By training against a diverse population of opponents, the agent optimizes against a variety of opponent strategies. Population-based training has shown success in pure coordination settings (Lupu et al., 2021) as well as completely adversarial games (Vinyals et al., 2019).

A second major barrier is reasoning with imperfect information. In partially observable environments (e.g., Poker), an agent needs to maintain its belief over world states (e.g., the hands of the opponents) during a planning procedure. Specific techniques such as counterfactual regret minimization (CFR) were developed (Zinkevich et al., 2008; Brown et al., 2020; Schmid et al., 2021b) for computer poker, where belief states can be characterized exactly (Moravčík et al., 2017). Exact reasoning about belief states can be intractable for domains with more complex forms of imperfect information, such as Stratego (Perolat et al., 2022). Approaches such as particle filtering may be applicable (Silver & Veness, 2010), but are also subject to scaling challenges.

We propose a general-purpose multiagent RL training regime to address the above issues, and extend AlphaZero-style RL and MCTS methods to large general-sum, imperfect information domains. The outer loop adopts a population-based training framework instantiated by *Policy Space Response Oracles* (PSRO) (Lanctot et al., 2017). PSRO incrementally generates a set of diverse opponents by repeating the following two steps: the *i) meta-strategy solver (MSS) step*, which computes a distribution over existing strategies via empirical game-theoretic analysis (EGTA) (Wellman, 2006), and *ii)* the *best response (BR) step*, which computes approximate best response policies using deep RL against the MSS distribution, adding them to the pool. This procedure effectively builds a belief hierarchy consisting of game-theoretic rationalizable strategies (Bernheim, 1984), bearing some resemblance to the $K$-level cognitive hierarchy (Camerer et al., 2004; Cui et al., 2021) of behavioral game theory and recursive reasoning in multiagent applications (Gmytrasiewicz & Durfee, 2000).

We employ an enhanced version of AlphaZero-style MCTS to train each best response strategy, thereby equipping our agent with the capability to both plan and infer the environmental state as well as opponents' strategic choices during online decision-making. This novel search method integrates deep RL with *Information Set MCTS* (IS-MCTS). To handle large imperfect information, we augment a deep generative model that samples world states at the root of the search tree, and iteratively refine its quality together with a PVN using RL trajectory data during the training loop. On each simulation step, a world state is sampled, and posterior mixed strategies of the opponents are updated, given the history implied by this world state. Then the opponent nodes are replaced with a sampled pure strategies from this distribution. Each pure strategy in the opponent pool serves as a "type" (Harsanyi, 1967) of play by viewing the environment as a Bayesian game. This type-based reasoning is also reflected by a recent work on Diplomacy (Bakhtin et al., 2023). While their work generates different types by sampling different regularization parameters of human policies, our approach automates the generations of types during the best response step, and calibrates the type distribution during the MSS step. Therefore, our agent is capable of performing test-time search while automatically inferring opponents' types given an observation history.

Experimentally, we first assess the capacity of PSRO to compute a Nash equilibrium across various benchmark games. We then test on two negotiation game domains: colored-trails and deal-or-no-deal. Our negotiation-based PSRO agents, selected using fairness criteria, reach Pareto frontier and achieve and a social welfare when negotiating with humans that is comparable to humans trading among themselves. Importantly, as was recently demonstrated in the cooperative game Overcooked (Strouse et al., 2021), this is achieved without using any human data in the training procedure.

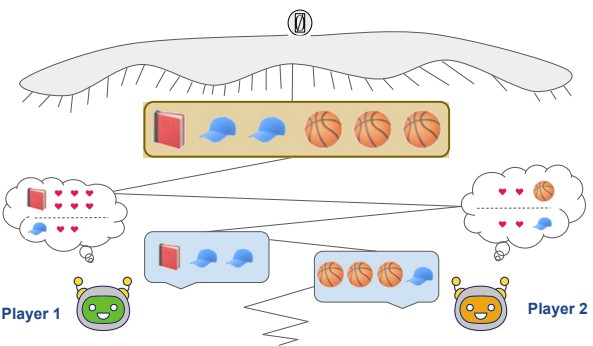

Figure 1: Example negotiation game in extensive-form. In "Deal or No Deal", the game starts at the empty history ($\emptyset$), chance samples a public pool of resources and private preferences for each player, then players alternate proposals for how to split the resources.

**Our Contributions:** We propose a general-purpose training regime for multiagent (partially observable) general sum, $n$-player, and negotiation games using game-theoretic RL. We make the following key extensions to AlphaZero and the PSRO framework:

- We integrate an AlphaZero-style Monte Carlo tree search (MCTS) approximate best response into the *best-response step* in PSRO. We further enhance this by incorporating deep-generative models into the best-response step training loop, which allows us to tractably represent belief-states during search in large imperfect information games (Section 3).

- We introduce and evaluate several new *meta-strategy solvers*, including those based on bargaining theory, which are particularly well suited for negotiation games (Section 4).

- We conduct an extensive evaluation across a variety of benchmark games (Appendix 5.1) and in two negotiation games, including one with human participants (Section 5).

## 2 Background and Related Work

An $n$-player **normal-form game** consists of a set of players $\mathcal{N} = \{1, 2, \ldots, n\}$, $n$ finite pure strategy sets $\Pi_i$ (one per player) with joint strategy set $\Pi = \Pi_1 \times \Pi_2 \times \cdots \Pi_n$, and a utility tensor (one per player), $u_i : \Pi \to \mathbb{R}$, and we denote player $i$'s utility as $u_i(\pi)$. Two-player (2P) normal-form games are called **matrix games**. A **two-player zero-sum** (purely adversarial) game is such that, $n = 2$ and for all joint strategies $\pi \in \Pi : \sum_{i \in \mathcal{N}} u_i(\pi) = 0$, whereas a **common-payoff** (purely cooperative) game: $\forall \pi \in \Pi, \forall i, j \in \mathcal{N} : u_i(\pi) = u_j(\pi)$. A **general-sum** game is one without any restrictions on the utilities. A **mixed strategy** for player $i$ is a probability distribution over $\Pi_i$ denoted $\sigma_i \in \Delta(\Pi_i)$, and a strategy profile $\sigma = \sigma_1 \times \cdots \times \sigma_n$, and for convenience we denote $u_i(\sigma) = \mathbb{E}_{\pi \sim \sigma}[u_i(\pi)]$. By convention, $-i$ refers to player $i$'s opponents. A **best response** is a strategy $b_i(\sigma_{-i}) \in \mathrm{BR}(\sigma_{-i}) \subseteq \Delta(\Pi_i)$, that maximizes the utility against a specific opponent strategy: for example, $\sigma_1 = b_1(\sigma_{-1})$ is a best response to $\sigma_{-1}$ if $u_1(\sigma_1, \sigma_{-1}) = \max_{\sigma_1'} u_1(\sigma_1', \sigma_{-1})$. An approximate $\boldsymbol{\epsilon}$**-Nash equilibrium** is a profile $\sigma$ such that for all $i \in \mathcal{N}, u_i(b_i(\sigma_{-i}), \sigma_{-i}) - u_i(\sigma) \leq \epsilon$, with $\epsilon = 0$ corresponding to an exact Nash equilibrium.

A "correlation device", $\mu \in \Delta(\Pi)$, is a distribution over the *joint* strategy space, which secretly recommends strategies to each player. Define $u_i(\pi_i', \mu)$ to be the expected utility of $i$ when it deviates to $\pi_i'$ given that other players follow their recommendations from $\mu$. Then, $\mu$ is **coarse-correlated equilibrium (CCE)** when no player $i$ has an incentive to unilaterally deviate *before* receiving their recommendation: $u_i(\pi_i', \mu) - u_i(\mu) \leq 0$ for all $i \in \mathcal{N}, \pi_i' \in \Pi_i$. Similarly, define $u_i(\pi_i', \mu|\pi_i'')$ to be the expected utility of deviating to $\pi_i'$ given that other players follow $\mu$ and player $i$ has received recommendation $\pi_i''$ from the correlation device. A **correlated equilibrium (CE)** is a correlation device $\mu$ where no player has an incentive to unilaterally deviate *after* receiving their recommendation: $u_i(\pi_i', \mu|\pi_i'') - u_i(\mu|\pi_i'') \leq 0$ for all $i \in \mathcal{N}, \pi_i' \in \Pi_i, \pi_i'' \in \Pi_i$.

---

**Algorithm 1** Policy-Space Response Oracles (PSRO)

---

**Input:** Game $\mathcal{G}$, Meta Strat. Solver MSS, oracle `BR`.
**function** PSRO($\mathcal{G}$, MSS, BR)
    Initialize strategy sets $\forall n, \Pi_i = \{\pi_i^0\}$. Initialize mixed strategies $\sigma_i(\pi_i^0) = 1, \forall i$, payoff tensor $U^0$.
    **for** $t \in \{0, 1, 2 \cdots, T\}$ **do**
      **for** $i \in \mathcal{N}$ **do**
        $\Pi_i \leftarrow \Pi_i \bigcup \{\text{BR}(i, \sigma, num\_eps)\}$
      **end for**
      Update missing entries in $U^t$ via simulations
      $\sigma \leftarrow \text{MSS}(U^t)$
    **end for**
    **return** $\Pi = (\Pi_1, \Pi_2, \cdots, \Pi_n), \sigma$
**end function**

---

In an **extensive-form game**, play takes place over a sequence of actions $a \in \mathcal{A}$. Examples of such games include chess, Go, and poker. An illustrative example of interaction in an extensive-form game is shown in Figure 1. A **history** $h \in \mathcal{H}$ is a sequence of actions from the start of the game taken by all players. Legal actions are at $h$ are denoted $\mathcal{A}(h)$ and the player to act at $h$ as $\tau(h)$. Players only partially observe the state and hence have imperfect information. There is a special player called **chance** that plays with a fixed stochastic policy (selecting outcomes that represent dice rolls or private preferences). Policies $\pi_i$ (also called behavioral strategies) is a collection of distributions over legal actions, one for each player's **information state**, $s \in \mathcal{S}_i$, which is a set of histories consistent with what the player knows at decision point $s$ (*e.g.* all the possible private preferences of other players), and $\pi_i(s) \in \Delta(\mathcal{A}(s))$.

There is a subset of the histories $\mathcal{Z} \subset \mathcal{H}$ called **terminal histories**, and utilities are defined over terminal histories, e.g. $u_i(z)$ for $z \in \mathcal{Z}$ could be –1 or 1 in Go (representing a loss and a win for player $i$, respectively). As before, expected utilities of a joint profile $\pi = \pi_1 \times \cdots \times \pi_n$ is defined as an expectation over the terminal histories, $u_i(\pi) = \mathbb{E}_{z \sim \pi}[u_i(z)]$, and best response and Nash equilibria are defined with respect to a player's full policy space.

## 2.1 EGTA and Policy-Space Response Oracles

Empirical game-theoretic analysis (EGTA) (Wellman, 2006) is an approach to reasoning about large sequential games through normal-form **empirical game** models, induced by simulating enumerated subsets of the players' full policies in the sequential game. Policy-Space Response Oracles (PSRO) (Lanctot et al., 2017) uses EGTA to incrementally build up each player's set of policies ("oracles") through repeated applications of approximate best response using RL. Each player's initial set contains a single policy (*e.g.* uniform random) resulting in a trivial empirical game $U^0$ containing one cell. On epoch $t$, given $n$ sets of policies $\Pi_i^t$ for $i \in \mathcal{N}$, utility tensors for the empirical game $U^t$ are estimated via simulation. A **meta-strategy solver** (MSS) derives a profile $\sigma^t$, generally mixed, over the empirical game strategy space. A new best response oracle, say $b_i^t(\sigma_{-i}^t)$, is then computed for each player $i$ by training against opponent policies sampled from $\sigma_{-i}^t$. These are added to strategy sets for the next epoch: $\Pi_i^{t+1} = \Pi_i^t \cup \{b_i^t(\sigma_{-i}^t)\}$. Since the opponent policies are fixed, the oracle response step is a single-agent problem (Oliehoek & Amato, 2014), and (deep) RL can feasibly handle large state and policy spaces.

### 2.1.1 Algorithms for Meta-Strategy Solvers

A key motivation for introducing the MSS abstraction in PSRO (Lanctot et al., 2017) was the observation that best-responding to exact Nash equilibrium tended to produce new policies overfit to the current solution. Abstracting the solver allows for consideration of alternative response targets, for example those that ensure continual training against a broader range of past opponents, and those that keep some lower bound probability $\gamma/|\Pi_i|$ of being selected.

The current work considers a variety of previously proposed MSSs: **uniform** (corresponding to fictitious play (Brown, 1951)), **projected replicator dynamics** (PRD), a variant of replicator dynamics with directed exploration (Lanctot et al., 2017), **$\alpha$-rank** (Omidshafiei et al., 2019; Muller et al., 2019), **maximum Gini (coarse) correlated equilibrium** (MGCE and MGCCE) solvers (Marris et al., 2021), and exploratory **regret-matching** (RM) (Hart & Mas-Colell, 2000), a parameter-free regret minimization algorithm commonly used in extensive-form imperfect information games (Zinkevich et al., 2008; Moravčík et al., 2017; Brown et al., 2020; Schmid et al., 2021a). We also use and evaluate ADIDAS (Gemp et al., 2021) as an MSS for the first time. ADIDAS is a recently proposed general approximate Nash equilibrium (limiting logit equilibrium / QRE) solver.

## 2.2 Combining MCTS and RL for Best Response

The performance of EGTA and PSRO depend critically on the quality of policies found in the best-response steps; to produce stronger policies and enable test-time search, AlphaZero-style combined RL+MCTS (Silver et al., 2018) can be used in place of the RL alone. This has been applied recently to find exploits of opponent policies in Approximate Best Response (ABR) (Timbers et al., 2022; Wang et al., 2023) and also combined with auxiliary tasks for opponent prediction in BRExIt (Hernandez et al., 2023). This combination can be particularly powerful; for instance, ABR found an exploit in a human-level Go playing agent trained with significant computational resources using AlphaZero.

When computing an approximate best response in imperfect information games, ABR uses a variant of Information Set Monte Carlo tree search (Cowling et al., 2012) called IS-MCTS-BR. At the root of the IS-MCTS-BR search (starting at information set $s$), the posterior distribution over world states, $\Pr(h \mid s, \pi_{-i})$ is computed explicitly, which requires both (i) enumerating every history in $s$, and (ii) computing the opponents' reach probabilities for each history in $s$. Then, during each search round, a world state is sampled from this belief distribution, then the game-tree regions are explored in a similar way as in the vanilla MCTS, and finally the statistics are aggregated on the information-set level. Steps (i) and (ii) are prohibitively expensive in games with large belief spaces. Hence, we propose learning a generative model online during the BR step; world states are sampled directly from the model given only their information state descriptions, leading to a succinct representation of the posterior capable of generalizing to large state spaces.

## 3 Search-Improved Generative PSRO

Our main algorithm has three components: the main driver (PSRO) (Lanctot et al., 2017), a search-enhanced BR step that concurrently learns a generative model, and the search with generative world state sampling itself. The main driver (Algorithm 1) operates as described in Subsection 2.1. In classical PSRO, the best response oracle is trained entirely via standard RL. For the first time, we introduce Approximate Best Response (ABR) as a search-based oracle in PSRO with a generative model for sampling world states.

The approximate best response step (Algorithm 2) proceeds analogously to AlphaZero's self-play based training, which trains a value net $\boldsymbol{v}$, a policy net $\boldsymbol{p}$, along with a generative network $\boldsymbol{g}$ using trajectories generated by search. There are some important differences from AlphaZero. Only one player is learning (*e.g.* player $i$). The (set of) opponents are fixed, sampled at the start of each episode from the opponent's meta-distribution $\sigma_{-i}$. Whenever it is player $i$'s turn to play, since we are considering imperfect information games, it runs a POMDP search procedure based on IS-MCTS (Algorithm 3) from its current information state $s_i$. The search procedure produces a policy target $\pi^*$, and an action choice $a^*$ that will be taken at $s_i$ at that episode. Data about the final outcome and policy targets for player $i$ are stored in data sets $D_{\boldsymbol{v}}$ and $D_{\boldsymbol{p}}$, which are used to improve the value net and policy net that guide the search. Data about the history, $h$, in each information set, $s(h)$, reached is stored in a data set $D_{\boldsymbol{g}}$, which is used to train the generative network $\boldsymbol{g}$ by supervised learning.

The MCTS search we use (Algorithm 3) is based on IS-MCTS-BR in (Timbers et al., 2022) (described in Section 2.2) and POMCP (Silver & Veness, 2010). Here it utilizes value net $\boldsymbol{v}$ to truncate the search at an unexpanded node and policy net $\boldsymbol{p}$ for action selection at an expanded node $s$ using the PUCT (Silver et al., 2018) formula: $\texttt{MaxPUCT}(s, \boldsymbol{p}) = \arg\max_{a \in \mathcal{A}(s)} \frac{s.child(a).value}{s.child(a).visits} + c_{uct} \cdot \boldsymbol{p}(s, a) \cdot \frac{\sqrt{s.total\_visits}}{s.child(a).visits + 1}$, for some

---

**Algorithm 2** ABR with generative model learning

---

**function** ABR($i, \sigma, num\_eps$)
    Initialize value nets $\boldsymbol{v}, \boldsymbol{v}'$, policy nets $\boldsymbol{p}, \boldsymbol{p}'$, generative nets $\boldsymbol{g}, \boldsymbol{g}'$, data buffers $D_{\boldsymbol{v}}, D_{\boldsymbol{p}}, D_{\boldsymbol{g}}$
    **for** $eps = 1, \ldots, num\_eps$ **do**
        $h \leftarrow$ initial state. $\mathcal{T} = \{s_i(h)\}$
        Sample opponents $\pi_{-i} \sim \sigma_{-i}$.
        **while** $h$ not terminal **do**
            **if** $\tau(h) = chance$ **then**
                Sample chance event $a \sim \pi_c$
            **else if** $\tau(h) \neq i$ **then**
                Sample $a \sim \pi_{\tau(h)}$
            **else**
                $a, \pi \leftarrow$ Search$(s_i(h), \sigma, \boldsymbol{v}', \boldsymbol{p}', \boldsymbol{g}')$
                $D_{\boldsymbol{p}} \leftarrow D_{\boldsymbol{p}} \bigcup \{(s_i(h), \pi)\}$
                $D_{\boldsymbol{g}} \leftarrow D_{\boldsymbol{g}} \bigcup \{(s_i(h), h)\}$
            **end if**
            $h \leftarrow h.apply(a)$, $\mathcal{T} \leftarrow \mathcal{T} \bigcup \{s_i(h)\}$
        **end while**
        $D_{\boldsymbol{v}} \leftarrow D_{\boldsymbol{v}} \bigcup \{(s, r) \mid s \in \mathcal{T}\}$, where $r$ is the payoff of $i$ in this trajectory
        $\boldsymbol{v}, \boldsymbol{p}, \boldsymbol{g} \leftarrow$ Update$(\boldsymbol{v}, \boldsymbol{p}, \boldsymbol{g}, D_{\boldsymbol{v}}, D_{\boldsymbol{p}}, D_{\boldsymbol{g}})$
        Replace parameters of $\boldsymbol{v}', \boldsymbol{p}', \boldsymbol{g}'$ by the latest parameters of $\boldsymbol{v}, \boldsymbol{p}, \boldsymbol{g}$ periodically.
    **end for**
    **return** Search$(\cdot, \sigma, \boldsymbol{v}, \boldsymbol{p}, \boldsymbol{g})$, or policy network $\boldsymbol{p}$, or greedy policy towards $\boldsymbol{v}$
**end function**

---

constant $c_{uct}$. Then at the end of the search call, it returns an action $a^*$ which receives the most visits at the root node, and a policy $\pi^*$ representing the action distribution of the search at the root node.

Algorithm 3 has two important differences from previous methods. Firstly, rather than computing exact posteriors, we use the deep generative model $\boldsymbol{g}$ learned in Algorithm 2 to sample world states. As such, this approach may be capable of scaling to large domains where previous approaches such as particle filtering (Silver & Veness, 2010; Somani et al., 2013) fail. Secondly, in the context of PSRO the imperfect information of the underlying POMDP consists of both (i) the actual world state $h$ and (ii) opponents' pure-strategy commitment $\pi_{-i}$. We make use of the fact $\Pr(h, \pi_{-i} \mid s, \sigma_{-i}) = \Pr(h \mid s, \sigma_{-i}) \Pr(\pi_{-i} \mid h, \sigma_{-i})$ such that we approximate $\Pr(h \mid s, \sigma_{-i})$ by $\boldsymbol{g}$ and compute $\Pr(\pi_{-i} \mid h, \sigma_{-i})$ exactly via Bayes' rule. Computing $\Pr(\pi_{-i} \mid h, \sigma_{-i})$ can be interpreted as doing inference over opponents' types (Albrecht et al., 2016; Kreps & Wilson, 1982; Hernandez-Leal & Kaisers, 2017; Kalai & Lehrer, 1993) or styles during play (Synnaeve & Bessiere, 2011; Ponsen et al., 2010).

### 3.1 Extracting a Final Agent at Test Time

How can a single decision-making agent be extracted from $\Pi$? The naive method samples $\pi_i \sim \sigma_i$ at the start of each episode, then follows $\sigma_i$ for the episode. The **self-posterior** method *resamples* a new oracle $\pi_i \in \Pi_i^T$ at information states each time an action or decision is requested at $s$. At information state $s$, the agent samples an oracle $\pi_i$ from the posterior over its own oracles: $\pi_i \sim \Pr(\pi_i \mid s, \sigma_i)$, using reach probabilities of its own actions along the information states leading to $s$ where $i$ acted, and then follows $\pi_i$. The self-posterior method is based on the equivalent behavior strategy distribution that Kuhn's theorem (Kuhn, 1953) derives from the mixed strategy distribution over policies. The **aggregate policy** method takes this a step further and computes the average (expected self-posterior) policy played at each information state, $\bar{\pi}_i^T(s)$, exactly (rather than via samples); it is described in detail in (Lanctot et al., 2017, Section E.3). The **rational planning** method enables decision-time search instantiated with the final oracles: it assumes the opponents at test time exactly match the $\sigma_{-i}$ of training time, and keeps updating the posterior $\Pr(h, \pi_{-i} \mid s, \sigma_{-i})$ during an online play. Whenever it needs to take an action at state $s$ at test time, it employs Algorithm 3 to

---

**Algorithm 3** IS-MCTS-BR with generative sampling

---

> **function** Search($s$, $\sigma$, $\boldsymbol{v}$, $\boldsymbol{p}$, $\boldsymbol{g}$)
>> **for** $iter = 1, \ldots, num\_sim$ **do**
>>> $\mathcal{T} = \{\}$
>>> Sample a world state (gen. model): $h \sim \boldsymbol{g}(h \mid s)$
>>> Sample an opponent profile using Bayes' rule: $\pi'_{-i} \sim \Pr(\pi_{-i} \mid h, \sigma_{-i})$. Replace opponent nodes with chance events according to $\pi'_{-i}$
>>> **while  do**
>>>> **if** $h$ is terminal **then**
>>>>> $r \leftarrow$ payoff of $i$. **Break**
>>>> **else if** $\tau(h) = chance$ **then**
>>>>> $a \leftarrow$ sample according to chance
>>>> **else if** $s_i(h)$ not in search tree **then**
>>>>> Add $s_i(h)$ to search tree.
>>>>> $r \leftarrow \boldsymbol{v}(s_i(h))$
>>>> **else**
>>>>> $a \leftarrow$ MaxPUCT($s_i(h), \boldsymbol{p}$)
>>>>> $\mathcal{T} \leftarrow \mathcal{T} \cup \{(s_i(h), a)\}$
>>>> **end if**
>>>> $h.apply(a)$
>>> **end while**
>>> **for** $(s, a) \in \mathcal{T}$ **do**
>>>> $s.child(a).visits \leftarrow s.child(a).visits + 1$
>>>> $s.child(a).value \leftarrow s.child(a).value + r$
>>>> $s.total\_visits \leftarrow s.total\_visits + 1$
>>> **end for**
>> **end for**
>> **return** action $a^*$ that receives max visits among children of $s$, and a policy $\pi^*$ that represents the visit frequency of children of $s$
> **end function**

---

search against this posterior. This method combines online Bayesian opponent modeling and search-based best response, which resembles the rational learning process (Kalai & Lehrer, 1993).

## 4    New Meta-Strategy Solvers

Recall from Section 2 that a meta-strategy solver (MSS) selects a strategy profile from the current empirical game for use as best-response target. This target can take the form of either: (i) $\mu$, a joint distribution over $\Pi$, or (ii) $(\sigma_1, \sigma_2, \ldots, \sigma_n)$, a set of (independent) distributions over $\Pi_i$, respectively. These distributions ($\mu_{-i}$ or $\sigma_{-i}$) are used to sample opponents when player $i$ is computing an approximate best response. We use many MSSs: several new and from previous work, summarized in Appendix A and Table 1. We present several new MSSs for general-sum games inspired by bargaining theory, which we now introduce.

### 4.1    Bargaining Theory and Solution Concepts

The **Nash Bargaining solution** (NBS) selects a Pareto-optimal payoff profile that uniquely satisfies axioms specifying desirable properties of invariance, symmetry, and independence of irrelevant alternatives (Nash, 1950; Ponsati & Watson, 1997). The axiomatic characterization of NBS abstracts away the process by which said outcomes are obtained through strategic interaction. However, Nash showed that it corresponds to a strategic equilibrium if threats are credible (Nash, 1953), and in fact, in bargaining games where agents take turns, under certain conditions the perfect equilibrium corresponds to the NBS (Binmore et al., 1986).

| Algorithm | Abbreviation | Independent/Joint | Solution Concepts | Description |
|---|---|---|---|---|
| $\alpha$-Rank | — | Joint | MCC | (Omidshafiei et al., 2019; Muller et al., 2019) |
| ADIDAS | — | Independent | LLE/QRE | (Gemp et al., 2021) |
| Max Entropy (C)CE | ME(C)CE | Joint | (C)CE | (Ortiz et al., 2007) |
| Max Gini (C)CE | MG(C)CE | Joint | (C)CE | (Marris et al., 2021) |
| Max NBS (C)CE | MN(C)CE | Joint | (C)CE | Sec 4.3 |
| Max Welfare (C)CE | MW(C)CE | Joint | (C)CE | (Marris et al., 2021) |
| Nash Bargaining Solution (NBS) | NBS | Independent | P-E | Sec 4.2 |
| NBS Joint | NBS_joint | Joint | P-E | Sec 4.2 |
| Projected Replicator Dynamics | PRD | Independent | ? | (Lanctot et al., 2017; Muller et al., 2019) |
| Regret Matching | RM | Independent | CCE | (Lanctot et al., 2017) |
| Social Welfare | SW | Joint | MW | Sec 4 |
| Uniform | — | Independent | ? | (Brown, 1951; Shoham & Leyton-Brown, 2009) |

Table 1: Meta-strategy solvers. For each MSS, we indicate whether its output is over joint or individual strategy spaces, and the solution concept it captures. P-E stands for Pareto efficiency.

---

**Algorithm 4** NBS by projected gradient ascent

---

**Input:** Initial iterate $\mathbf{x}$, payoff tensor $U$.
**function** NBS($\mathbf{x}^0$, $U$)
    Let $g(\mathbf{x})$ be the log Nash product defined in eqn (2)
    **for** $t = 0, 1, 2 \cdots, T$ **do**
        $\mathbf{y}^{t+1} \leftarrow \mathbf{x}^t + \alpha^t \nabla g(\mathbf{x}^t)$
        $\mathbf{x}^{t+1} \leftarrow \texttt{Proj}(\mathbf{y}^{t+1})$
    **end for**
    **Return** $\arg\max_{\mathbf{x}^{t=0:T}} g(\mathbf{x}^t)$
**end function**

---

Define the set of achievable payoffs as all expected utilities $u_i(\mu)$ under a joint-policy profile $\mu$ (Harsanyi & Selten, 1972; Morris, 2012). Denote the disagreement outcome of player $i$, which is the payoff it gets if no agreement is achieved, as $d_i$. The NBS is the set of policies that maximizes the **Nash bargaining score** (A.K.A. *Nash product*):

$$\max_{\mu \in \Delta(\Pi)} \Pi_{i \in \mathcal{N}} \left( u_i(\mu) - d_i \right), \tag{1}$$

which, when $n = 2$, leads to a quadratic program (QP) with the constraints derived from the policy space structure (Griffin, 2010). However, even in this simplest case of two-player matrix games, the objective is non-concave posing a problem for most QP solvers. Furthermore, scaling to $n$ players requires higher-order polynomial solvers.

## 4.2 Empirical Game Nash Bargaining Solution

Instead of using higher-order polynomial solvers, we propose an algorithm based on (projected) gradient ascent (Singh et al., 2000; Boyd & Vandenberghe, 2004). Let $\mathbf{x} \in \Delta(\Pi)$ represent a distribution over joint strategies in an empirical game. Let $u_i(\mathbf{x}) = \mathbb{E}_{\pi \sim \mathbf{x}}[u_i(\pi)]$ be the expected utility for player $i$ under the joint distribution $\mathbf{x}$. Let $\Pi_{i \in \mathcal{N}}(u_i(\mathbf{x}) - d_i)$ be the Nash product defined in Equation 1. In practice, $d_i$ is either clearly defined from the context, or is set as a value that is lower than the minimum achievable payoff of $i$ in $\Delta(\Pi)$. We restrict $u_i(\mathbf{x}) - d_i > 0$ for all $i, \mathbf{x}$. Note that the Nash product is non-concave, so instead of maximizing it, we maximize the **log Nash product** $g(\mathbf{x}) =$

$$\log \left( \Pi_{i \in \mathcal{N}}(u_i(\mathbf{x}) - d_i) \right) = \sum_{i \in \mathcal{N}} \log(u_i(\mathbf{x}) - d_i), \tag{2}$$

which has the same maximizers as equation 1, and is a sum of concave functions, hence concave. The process is depicted in Algorithm 4; `Proj` is the $\ell_2$ projection onto the simplex.

**Theorem 4.1.** *Assume any deal is better than no deal by $\kappa > 0$, i.e., $u_i(\mathbf{x}) - d_i \geq \kappa > 0$ for all $i, \mathbf{x}$. Let $\{\mathbf{x}^t\}$ be the sequence generated by Algorithm 4 with starting point $\mathbf{x}^0 = |\Pi|^{-1}\mathbf{1}$ and step size sequence*

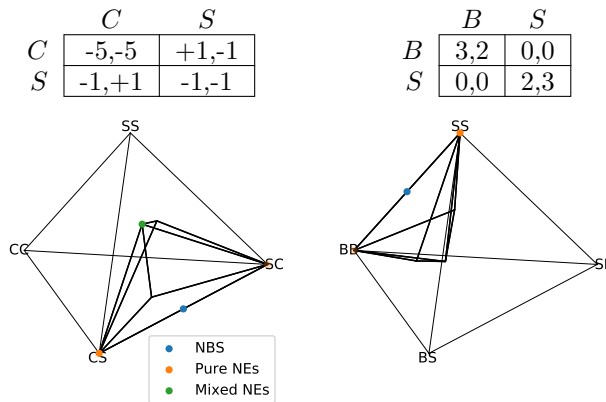

Figure 2: (C)CE polytopes in Chicken (left) and Bach-or-Stravinsky (right) showing NBS equilibrium selection.

$\alpha^t = \frac{\kappa\sqrt{(|\Pi|-1)/|\Pi|}}{u^{\max}n}(t+1)^{-1/2}$. *Then, for all $t > 0$ one has*

$$\max_{\mathbf{x}\in\Delta^{|\Pi|-1}} g(\mathbf{x}) - \max_{0\leq s\leq t} g(\mathbf{x}^s) \leq \frac{u^{\max}n\sqrt{|\Pi|}}{\kappa\sqrt{t+1}} \tag{3}$$

*where $u^{\max} = \max_{i,\mathbf{x}} u_i(\mathbf{x})$, $|\Pi|$ is the number of possible pure joint strategies, and $\mathbf{x}$ is assumed to be a joint correlation device ($\mu$).*

For a proof, see Appendix B.

### 4.3 Max-NBS (Coarse) Correlated Equilibria

The second new MSS we propose uses NBS to select a (C)CE. For all normal-form games, valid (C)CEs are a convex polytope of joint distributions in the simplex defined by the linear constraints. Therefore, maximizing any strictly concave function can uniquely select an equilibrium from this space (for example Maximum Entropy (Ortiz et al., 2007), or Maximum Gini (Marris et al., 2021)). The log Nash product (Equation equation 2) is concave but not, in general, strictly concave. Therefore to perform unique equilibrium selection, a small additional strictly concave regularizer (such as maximum entropy) may be needed to select a uniform mixture over distributions with equal log Nash product. We use existing off-the-shelf exponential cone constraints solvers (e.g., ECOS (Domahidi et al., 2013), SCS (O'Donoghue et al., 2021)) which are available in simple frameworks (CVXPY (Diamond & Boyd, 2016; Agrawal et al., 2018)) to solve this optimization problem.

NBS is a particularly interesting selection criterion. Consider the game of chicken in where players are driving head-on; each may continue (C), which may lead to a crash, or swerve (S), which may make them look cowardly. Many joint distributions in this game are valid (C)CE equilibria (Figure 2). The optimal outcome in terms of both welfare and fairness is to play SC and CS each 50% of the time. NBS selects this equilibrium. Similarly in Bach-or-Stravinsky where players coordinate but have different preferences over events: the fairest maximal social welfare outcome is a compromise, mixing equally between BB and SS.

### 4.4 Social Welfare

This MSS selects the pure joint strategy of the empirical game that maximizes the estimated social welfare.

## 5 Experiments

We initially assessed PSRO as a general approximate Nash equilibrium solver and collection of MSSs over 12 benchmark games commonly used throughout the literature. The full results are presented in Appendix 5.1.

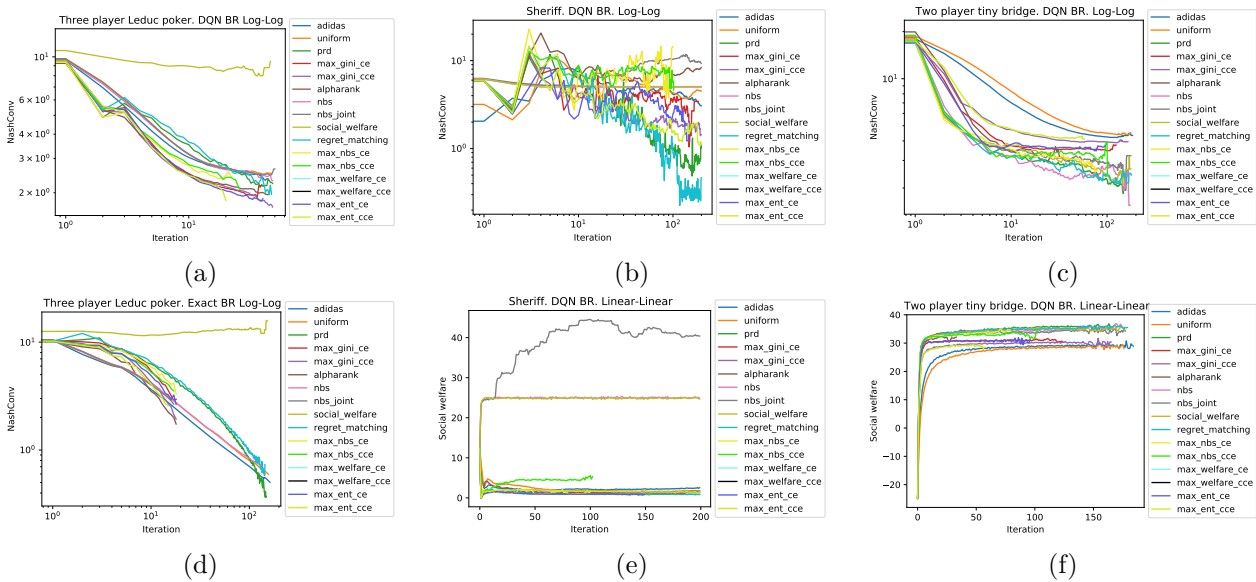

Figure 3: NASHCONV and social welfare along PSRO iterations across game types. NASHCONV in 3P Leduc poker using (a) DQN oracles vs. (d) exact oracles. NASHCONV (b) and (e) social welfare in Sheriff. NASHCONV (c) and social welfare (f) in 2P Tiny Bridge.

Then we focus our evaluation on negotiation, a common human interaction and important class of general-sum games with a tension between incentives (competing versus cooperating).

## 5.1 Approximate Nash Equilibrium Solving on Benchmark Games

Here we evaluate the capacity of PSRO with different meta-strategy solvers [1] to act as a general Nash equilibrium solver for sequential $n$-player games. For these initial experiments, we run PSRO on its own without search. Since these are benchmark games, they are small enough to compute exact exploitability, or the Nash gap (called NASHCONV by Lanctot et al. (2017; 2019)), and search is not necessary. We run PSRO using 16 different meta-strategy solvers across 12 different benchmark games (three 2P zero-sum games, three $n$-player zero-sum games, two common-payoff games, and four general-sum games): instances of Kuhn and Leduc poker, Liar's dice, Trade Comm, Tiny Bridge, Battleship, Goofspiel, and Sheriff. These games have recurred throughout previous work, so we describe them in Appendix C; they are all partially-observable and span a range of different types of games.

A representative sample of the results is shown in Figure 3, whereas all the results are shown below. $\text{NASHCONV}(\bar{\pi}) = \sum_{i \in \mathcal{N}} u_i(b_i(\bar{\pi}_{-i}), \bar{\pi}_{-i}) - u_i(\bar{\pi})$, where $b_i$ is an exact best response, and $\bar{\pi}$ is the exact average policy using the aggregator method described in Section 3.1. A value of zero means $\bar{\pi}$ is Nash equilibrium, and values greater than zero correspond to an gap from Nash equilibrium. Social welfare is defined as $\text{SW}(\bar{\pi}) = \sum_i u_i(\bar{\pi})$.

Most of the meta-strategy solvers seem to reduce NASHCONV faster than a completely uninformed meta-strategy solver (uniform) corresponding to fictitious play, validating the EGTA approach taken in PSRO. In three-player (3P) Leduc, the NashConv achieved for the exact best response is an order of magnitude smaller than when using DQN. ADIDAS, regret-matching, and PRD are a good default choice of MSS in competitive games. The correlated equilibrium meta-strategy solvers are surprisingly good at reducing NASHCONV in the competitive setting, but can become unstable and even fail when the empirical game becomes large in the exact case. In the general-sum game Sheriff, the reduction of NASHCONV is noisy, with several of the

---

[1]For simplicity we use the same MSS for both guiding the BR step (strategy exploration) and evaluating NASHCONV at each iteration. Firmer conclusions about relative effectiveness for strategy exploration would require defining a fixed solver for evaluation purposes (Wang et al., 2022).

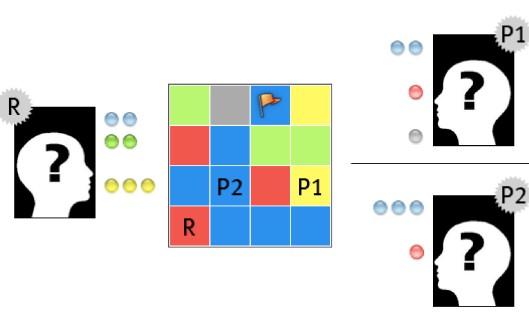

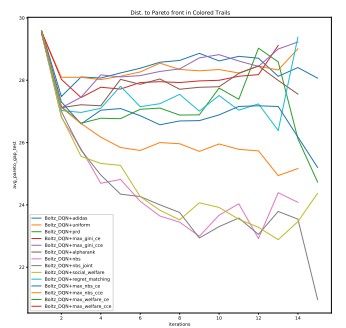

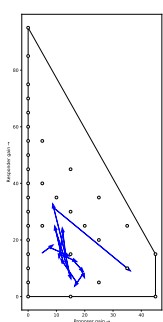

Figure 4: Three-Player Colored Trails.

Figure 5: Empirical reduction in Pareto Gap on test game configurations, and example evolution toward Pareto front (right).

meta-strategy solvers having erratic graphs, with RM and PRD performing best. Also in Sheriff, the Nash bargaining (and social welfare) meta-strategy solvers achieve significantly higher social welfare than most meta-strategy solvers. Similarly in the cooperative game of Tiny Bridge, the MSSs that reach closest to optimal are NBS (independent and joint), social welfare, RM, PRD, and Max-NBS-(C)CE. Many of these meta-strategy solvers are not *guaranteed* to compute an approximate Nash equilibrium (even in the empirical game), but any limiting logit equilibrium (QRE) solver can get arbitrarily close. ADIDAS does not require the storage of the meta-tensor $U^t$, only samples from it. So, as the number of iterations grow ADIDAS might be one of the safest and most memory-efficient choice for reducing NASHCONV long-term.

## 5.2 Negotiation Game: Colored Trails

We start with a highly configurable negotiation game played on a grid (Gal et al., 2010a) of colored tiles, which has been actively studied by the AI community (Grosz et al., 2004; Ficici & Pfeffer, 2008b; Gal et al., 2010b). Colored Trails does not require search since the number of moves is small, so we use classical RL based oracles (DQN and Boltzmann DQN) to isolate the effects of the new meta-strategy solvers. Furthermore, it has a property that most benchmark games do not: it is parameterized by a board (tile layout and resource) configuration, which allows for a training/testing set split to evaluate the capacity to generalize across different instances of similar games.

We use a three-player variant (Ficici & Pfeffer, 2008a; de Jong et al., 2011) depicted in Figure 4. At the start of each episode, an instance (a board and colored chip allocation per player) is randomly sampled from a database of strategically interesting and balanced configurations (de Jong et al., 2011, Section 5.1). There are two proposers (P1 and P2) and a responder (R). R can see all players' chips, both P1 and P2 can see R's chips; however, proposers cannot see each other's chips. Each proposer, makes an offer to the receiver. The receiver than decides to accept one offer and trades chips with that player, or passes. Then, players spend chips to get as close to the flag as possible (each chip allows a player to move to an adjacent tile if it is the same color as the chip). For any configuration (player $i$ at position $p$), define $\text{SCORE}(p, i) = (-25)d + 10t$, where $d$ is the Manhattan distance between $p$ and the flag, and $t$ is the number of player $i$'s chips. The utility for player $i$ is their *gain*: score at the end of the game minus the score at the start.

This game has been decomposed into specific hand-crafted meta-strategies for both proposers and receiver (de Jong et al., 2011). These meta-strategies cover the Pareto-frontier of the payoff space by construction. Rather than relying on domain knowledge, we evaluate the extent to which PSRO can learn such a subset of representative meta-strategies. To quantify this, we compute the Pareto frontier for a subset of configurations, and define the Pareto Gap (P-Gap) as the minimal $\ell_2$ distance from the outcomes to the outer surfaces of the convex hull of the Pareto front, which is then averaged over the set of configurations in the database.

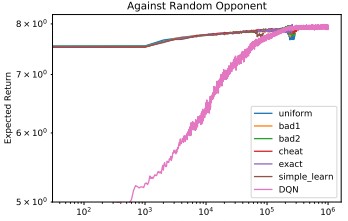 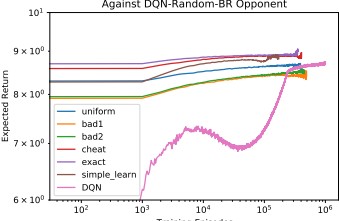 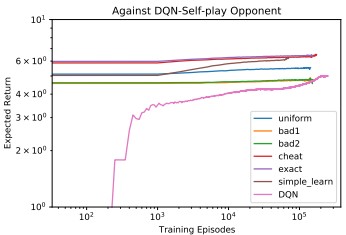

Figure 6: Best response performance using different generative models, against (left) uniform random opponent, (middle) DQN response to uniform random, (right) self-play DQN opponent. **Uniform** samples a legal preference vector uniformly at random, **bad1** always samples the first legal instance in the database, **bad2** always samples the last legal instance in the database, **cheat** always samples the actual underlying world state, **exact** samples from the exact posterior, **simple learn** is the method described in Algorithm 2 (detailed in Appendix D.1), and **DQN** is a simple DQN responder that does not use a generative model nor search.

Figure 5 shows representative results of PSRO agents on Colored Trails (for full graphs, and evolution of score diagrams, see Appendix E.1). The best-performing MSS is NBS-joint, beating the next best by a full 3 points. The NBS meta-strategy solvers comprise five of the six best MSSs under this evaluation. An example of the evolution of the expected score over PSRO iterations is also shown, moving toward the Pareto front, though not via a direct path.

### 5.3 Negotiation Game: Deal or No Deal

"Deal or No Deal" (DoND) is a simple alternating-offer bargaining game with incomplete information, which has been used in many AI studies (DeVault et al., 2015; Lewis et al., 2017; Cao et al., 2018; Kwon et al., 2021). Our focus is to train RL agents to play against humans *without human data*, similar to previous work (Strouse et al., 2021). An example game of DoND is shown in Figure 1. Two players are assigned *private* preferences $\mathbf{v}_1 \geq \mathbf{0}, \mathbf{v}_2 \geq \mathbf{0}$ for three different items (books, hats, and basketballs). At the start of the game, there is a pool $\mathbf{p}$ of 5 to 7 items drawn randomly such that: (i) the total value for a player of all items is 10: $\mathbf{v}_1 \cdot \mathbf{p} = \mathbf{v}_2 \cdot \mathbf{p} = 10$, (ii) each item has non-zero value for at least one player: $\mathbf{v}_1 + \mathbf{v}_2 > \mathbf{0}$, (iii) some items have non-zero value for both players, $\mathbf{v}_1 \odot \mathbf{v}_2 \neq \mathbf{0}$, where $\odot$ represents element-wise multiplication.

The players take turns proposing how to split the pool of items, for up to 10 turns (5 turns each). If an agreement is not reached, the negotiation ends and players both receive 0. Otherwise, the agreement represents a split of the items to each player, $\mathbf{p}_1 + \mathbf{p}_2 = \mathbf{p}$, and player $i$ receives a utility of $\mathbf{v}_i \cdot \mathbf{p}_i$. DoND is an imperfect information game because the other player's preferences are private. We use a database of 6796 bargaining instances made publicly available in (Lewis et al., 2017). Deal or No Deal is a significantly large game, with an estimated $1.32 \cdot 10^{13}$ information states for player 1 and $5.69 \cdot 10^{11}$ information states for player 2 (see Appendix C.2 for details).

#### 5.3.1 Generative World State Sampling

We now show that both the search and the generative model contribute to achieving higher reward (in the BR step of PSRO) than RL alone. The input of our deep generative model is one's private values $\mathbf{v}_i$ and public observations, and the output is a distribution over $\mathbf{v}_{-i}$ (detailed in the Appendix). We compute approximate best responses to three opponents: uniform random, a DQN agent trained against uniform random, and a DQN agent trained in self-play. We compare different world state sampling models as well to DQN in Figure 6, where the deep generative model approach is denoted as simple_learn.

The benefit of search is clear: the search methods achieve a high value in a few episodes, a level that takes DQN many more episodes to reach (against random and DQN response to random) and a value that is not reached by DQN against the self-play opponent. The best generative models are the true posterior (exact) and the actual underlying world state (cheat). However, the exact posterior is generally intractable and the

| Name | Values |
|------|--------|
| Individual return (IR) | $r_i$ |
| Inequity aversion (Fehr & Schmidt, 1999) (IE) | $r_i - 0.5 \cdot \max\{r_{-i} - r_i, 0\}$ (Gal et al., 2010a) |
| Social welfare (SW) | $r_i + r_{-i}$ |
| Nash bargaining score (NBS) | $r_i \cdot r_{-i}$ |

Table 2: Different tree back-propagation value types. $r_i$ is the return for player $i$.

underlying world state is not accessible to the agent at test-time, so these serve as idealistic upper-bounds. Uniform seems to be a compromise between the bad and ideal models. The deep generative model approach is roughly comparable to uniform at first, but learns to approximate the posterior as well as the ideal models as data is collected. In contrast, DQN eventually reaches the performance of the uniform model against the weaker opponent but not against the stronger opponent even after 20000 episodes.

### 5.3.2 Studies with Human Participants

We recruited participants from Prolific (Pe'er et al., 2021; Peer et al., 2017) to evaluate the performance of our agents in DoND (overall $N = 346$; 41.4% female, 56.9% male, 0.9% trans or nonbinary; median age range: 30–40). Crucially, participants played DoND for real monetary stakes, with an additional payout for each point earned in the game.

Participants first read game instructions and completed a short comprehension test to ensure they understood key aspects of DoND's rules (see Appendix F for instruction text and study screenshots). Participants then played five episodes of DoND with a randomized sequence of opponents. Episodes terminated after players reached a deal, after 10 moves without reaching a deal, or after 120 seconds elapsed. After playing all five episodes, participants completed a debrief questionnaire collecting standard demographic information and open-ended feedback on the study. See Appendix F for additional details.

**Training Details** Our infrastructure restricts that each human participant can only play five matches with our bots. Therefore we decided to select five different agents so every participant can play each of these once. For comparison, we decided to include one independent RL agent and four search-improved PSRO agents of different playing styles.

For the independent RL agent, we trained two classes of independent RL agents in selfplay: (1) DQN (Mnih et al., 2015) and Boltzmann DQN (Cui & Koeppl, 2021), and (2) policy gradient algorithms such as A2C, QPG, RPG and RMPG (Srinivasan et al., 2018). For DQN and Boltzmann DQN, we used replay buffer sizes of $10^5$, $\epsilon$ decayed from $0.9 \rightarrow 0.1$ over $10^6$ steps, a batch size of 128, and swept over learning rates of $\{0.01, 0.02, 0.005\}$. For Boltzmann DQN, we varied the temperature $\eta \in 0.25, 0.5, 1$. For all self-play policy gradient methods we used a batch size of 128, and swept over critic learning rate in $\{0.01, 0.001\}$, policy learning rate in $\{.001, 0.0005, 0.0001\}$, number of updates to the critic before updating the policy in $\{1, 4, 8\}$, and entropy cost in $\{0.01, 0.001\}$. DQN trained with the settings above and a learning rate of 0.005 was the agent we found to achieve highest individual returns (and social welfare, and Nash bargaining score), so we select it as the representative agent for the independent RL category.

For PSRO agents, we consider 16 different meta-strategy solvers, and 4 different back-propagating value types during the tree search procedure, making it 64 different combination in total. Notice that the original MCTS algorithm (Algorithm 3) back-propagates individual rewards during each simulation phase for the search agent. We also explore other choices such as social welfare and inequity aversion in our DoND human behavioral studies, as listed in Table 2.

We consider self-posterior (SP) and rational planning (RP) methods described in Section 3.1 to extract a final decision agent, as the other approaches are either infeasible computationally or is subsumed by the current methods. That makes it 128 agents totally in principle. We train the neural networks for 3 days, and screen out those combination that cannot make it to the 3rd PSRO iteration (due to break of the MSS optimizer). We eventually have 112 different PSRO agents at hand. As detailed in App. E.2, we apply empirical game-theoretic analysis on the resulting $112 \times 112$ meta-game and using different categories

| Agent | $\bar{u}_{\text{Humans}}$ | | $\bar{u}_{\text{Agent}}$ | | $\bar{u}_{\text{Comb}}$ | | NBS |
|---|---|---|---|---|---|---|---|
| IndRL | 5.86 | $[5.37, 6.40]$ | **6.50** | $[\mathbf{5.93}, \mathbf{7.06}]$ | 6.18 | $[5.82, 6.56]$ | 38.12 |
| Comp1 | 5.14 | $[4.56, 5.63]$ | 5.49 | $[4.87, 6.11]$ | 5.30 | $[4.93, 5.76]$ | 28.10 |
| Comp2 | 6.00 | $[5.49, 6.55]$ | 5.54 | $[4.96, 6.10]$ | 5.76 | $[5.33, 6.12]$ | 33.13 |
| Coop | 6.71 | $[6.23, 7.20]$ | 6.17 | $[5.66, 6.64]$ | 6.44 | $[6.11, 6.75]$ | 41.35 |
| Fair | **7.39** | $[\mathbf{6.89}, \mathbf{7.87}]$ | 5.98 | $[5.44, 6.49]$ | **6.69** | $[\mathbf{6.34}, \mathbf{7.01}]$ | **44.23** |

Table 3: Humans vs. agents performance with $N = \mathbf{129}$ human participants, **547** games total. $\bar{u}_X$ refers to the average utility to group $X$ (for the humans when playing the agent, or for the agent when playing the humans), Comb refers to Combined (human and agent). Square brackets indicate 95% confidence intervals. IndRL refers to Independent RL (DQN), Comp1 and Comp2 are the two top-performing competitive agents, Coop is the most cooperative agent, and Fair is fairest agent. NBS is the Nash bargaining score (Eq 1).

to select the final four PSRO agents. We eventually selected: (i) two most competitive agents (Comp1, Comp2) (maximizing utility), (ii) the most cooperative agents (Coop) (maximizing social welfare), the (iii) the fairest agent (Fair) (minimizing social inequity (Fehr & Schmidt, 1999)); (iv) a separate top-performing independent RL agent (IndRL) trained in self-play (DQN). Here Comp1, Comp2 are extracted using SP while Coop and Fair are using RP. Both Coop and Fair are using Nash product as the back-propagating values during tree search, while Comp1 uses inequity aversion and Comp2 uses individual rewards. Comp1, Comp2 and Fair are trained using Max-Gini CE or CCE MSS, while Coop uses uniform distribution as the MSS.

**Results** We collect data under two conditions: human vs. human (HvH), and human vs. agent (HvA). In the HvH condition, we collect 483 games: 482 end in deals made (99.8%), and achieve a return of 6.93 (95% c.i. $[6.72, 7.14]$), on expectation. We collect 547 games in the HvA condition: 526 end in deals made (96.2%; see Table 3). DQN achieves the highest individual return. By looking at the combined reward, it achieves this by aggressively reducing the human reward (down to 5.86)–possibly by playing a policy that is less human-compatible. The competitive PSRO agents seem to do the same, but without overly exploiting the humans, resulting in the lowest social welfare overall. The cooperative agent achieves significantly higher combined utility playing with humans. Better yet is Human vs. Fair, the only Human vs. Agent combination to achieve social welfare comparable to the Human vs. Human social welfare.

Another metric is the objective value of the empirical NBS from Eq. 1, over the symmetric game (randomizing the starting player) played between the different agent types. This metric favors Pareto-efficient outcomes, balancing between the improvement obtained by both sides. From App E.2, the NBS of Coop decreases when playing humans, from $44.51 \rightarrow 41.35$– perhaps due to overfitting to agent-learned conventions. Fair increases slightly ($42.56 \rightarrow 44.23$). The NBS of DQN rises from $23.48 \rightarrow 38.12$. The NBS of the competitive agents also rises playing against humans ($24.70 \rightarrow 28.10$, and $25.44 \rightarrow 28.10$), and also when playing with Fair ($24.70 \rightarrow 29.63$, $25.44 \rightarrow 28.73$).

The fair agent is both adaptive to many different types of agents, and cooperative, increasing the social welfare in all groups it negotiated with. This could be due to its MSS (MGCE) putting significant weight on many policies leading to Bayesian prior with high support, or its backpropagation of the product of utilities rather than individual return.

## 6 Conclusion and Future Work

We proposed a general-purpose multiagent training regime that combines the power of MCTS search and a population-based training framework, for general-sum imperfect information domains. We developed a novel search technique that combines IS-MCTS with a deep belief learning module coupled with the RL training loop, which scale to large belief and state spaces. The outer loop of our algorithm is implemented by PSRO, which iteratively trains and adds search strategies guided by game-theoretic analysis. On one hand, search serves as a strong best response method within the PSRO loop, which provides an instance of the framework of its own interests. On the other hand, PSRO automatically produces a belief hierarchy over the opponents'

strategies, which endows the search with the capability of inferring opponent types during online decision makings. This dual view of the whole training architecture illustrates its effectiveness in producing agents that are capable of opponent modeling through game-theoretic analysis and planning forward at test-time.

Our experimental results found that ADIDAS, regret-matching, and PRD MSSs work well generally and even better in competitive games. In cooperative, general-sum games and negotiation games, NBS-based meta-strategy solvers can increase social welfare and find solutions closer to the Pareto frontier. In our human-agent study of a negotiation game, self-play DQN exploits humans most, and agents trained with PSRO (selected using fairness criteria) adapt and cooperate well with humans.

Future work could further enhance the best response by predictive losses (Hernandez et al., 2023), scale to even larger domains, and convergence to other solution concepts such as self-confirming or correlated equilibria.

## 7    Broader Impact

We believe our search-improved PSRO method advances the general equilibrium solving and planning techniques in multi-agent systems with little domain knowledge. Our methods can be potentially deployed in a variety of applications, including automated bidding in auctions, negotiation, cybersecurity, warehouse robotics, and autonomous vehicle systems. All of these are multi-agent scenarios that involve general-sum, imperfect information elements.

One of the potential risks is value misalignment in negotiation. The method can produce strategies that are unpredictable and not easily explained, which could lead to exploitative behaviors in negotiation that are misaligned with the users' intents. This could potentially cause harm in the economic system and reduce market efficiency. Any deployed use of artificial agents built using our algorithm would need to first be thoroughly tested, ideally by third party, and undergo a controlled private study with humans to identify any potentially harmful behavior.

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

# A   Meta-Strategy Solvers

In this section, we describe the algorithms used for the MSS step of PSRO, which computes a set of meta-strategies $\sigma_i$ or a correlation device $\mu$ for the normal-form empirical game.

## A.1   Classic PSRO Meta-Strategy Solvers

### A.1.1   Projected Replicator Dynamics (PRD)

In the replicator dynamics, each player $i$ used by mixed strategy $\sigma_i^t$, often interpreted as a distribution over population members using pure strategies. The continuous-time dynamic then describes a change in weight on strategy $\pi_k \in \Pi_i$ as a time derivative:

$$\frac{d\sigma_i^t(\pi_k)}{dt} = \sigma_i^t(\pi_k)[u_i(\pi_k, \sigma_{-i}) - u_i(\sigma)].$$

The projected variant ensures that the strategies stay within the exploratory simplex such $\sigma_i^t$ remains a probability distribution, and that every elements is subject to a lower-bound $\frac{\gamma}{|\Pi_i|}$. In practice, this is simulated by small discrete steps in the direction of the time derivatives, and then projecting $\sigma_i^t$ back to the nearest point in the exploratory simplex.

### A.1.2   Exploratory Regret-Matching

Regret-Matching is based on the algorithm described in (Hart & Mas-Colell, 2000) and used in extensive-form games for regret minimization (Zinkevich et al., 2008). Regret-matching is an iterative solver that tabulates cumulative regrets $R_i^T(\pi)$, which initially start at zero. At each trial $t$, player $i$ would receive $u_i(\sigma_i^t, \sigma_{-i}^t)$ by playing their strategy $\sigma_i^t$. The instantaneous regret of *not* playing pure strategy $\pi_k \in \Pi_i$ is

$$r^t(\pi_k) = u_i(\pi_k, \sigma_{-i}^t) - u_i(\sigma_i^t, \sigma_{-i}^t).$$

The cumulative regret over $T$ trials for the pure strategy is then defined to be:

$$R_i^T(\pi_k) = \sum_{t=1}^{T} r^t(\pi_k).$$

Define $(x)^+ = \max(0, x)$. The policy at time $t + 1$ is derived entirely by these regrets:

$$\sigma_{i,RM}^{t+1}(\pi_k) = \frac{R_i^{T,+}(\pi_k)}{\sum_{\pi_k' \in \Pi_i} R_i^{T,+}(\pi_k')},$$

if the denominator is positive, or $\frac{1}{|\Pi_i|}$ otherwise. As in original PSRO, in this paper we also add exploration:

$$\sigma_i^{t+1} = \gamma \mathrm{UNIFORM}(\Pi_i) + (1 - \gamma)\sigma_{i,RM}^{t+1}.$$

Finally, the meta-strategy returned for all players at time $t$ is their average strategy $\bar{\sigma}^T$.

## A.2   Joint and Correlated Meta-Strategy Solvers

The jointly correlated meta-strategy solvers were introduced in (Marris et al., 2021), which was the first to propose computing equilibria in the joint space of the empirical game. In general, a correlated equilibrium (and coarse-correlated equilibrium) can be found by satisfying a number of constraints, on the correlation device, so the question is what to use as the optimization criterion, which effectively selects the equilibrium.

One that maximizes Shannon entropy (ME(C)CE) seems like a good choice as it places maximal weight on all strategies, which could benefit PSRO due to added exploration among alternatives. However it was found

to be slow on large games. Hence, Marris et al. (Marris et al., 2021) propose to use a different but related measure, the Gini impurity, for correlation device $\mu$,

$$\text{GINIIMPURITY}(\mu) = 1 - \mu^T \mu,$$

which is a form of Tsallis entropy. The resulting equilibria Maximum Gini (Coarse) Correlated Equilibria (MG(C)CE) have linear constraints and can be computed by solving a quadratic program. Also, Gini impurity has similar properties to Shannon entropy: it is maximized at the uniform distribution and minimized when all the weight is placed on a single element.

In the Deal-or-no-Deal experiments, 4 of 5 selected winners of tournaments used MG(C)CE (see Section E.2, including the one that cooperated best with human players, and the other used uniform. Hence, it is possible that the exploration motivated my high-support meta-strategies indeed does help when playing against a population of agents, possibly benefiting the Bayesian inference implied by the generative model and resulting search.

### A.3  ADIDAS

Average Deviation Incentive with Adaptive Sampling (ADIDAS) (Gemp et al., 2021) is an algorithm designed to approximate a particular Nash equilibrium concept called the *limiting logit equilibrium* (LLE). The LLE is unique in almost all $n$-player, general-sum, normal-form games and is defined via a homotopy (McKelvey & Palfrey, 1995). Beginning with a quantal response (logit) equilibrium (QRE) under infinite temperature (known to be the uniform distribution), a continuum of QREs is then traced out by annealing the temperature. The LLE, aptly named, is the QRE defined in the limit of zero temperature. ADIDAS approximates this path in a way that avoids observing or storing the entire payoff tensor. Instead, it intelligently queries specific entries from the payoff tensor by leveraging techniques from stochastic optimization.

## B  Nash Bargaining Solution of Normal-form games via Projected Gradient Ascent

In this section, we elaborate on the method proposed in Section 4.2. As an abuse of notation, let $u_i$ denote the payoff tensor for player $i$ flattened into a vector; similarly, let $\mathbf{x}$ be a vector as well. Let $d$ be the number of possible joint strategies, e.g., $m^n$ for a game with $n$ agents, each with $m$ pure strategies. Let $\Delta^{d-1}$ denote the $(d-1)$-simplex, i.e., $\sum_{k=1}^{d} \mathbf{x}_k = 1$ and $\mathbf{x}_k \geq 0$ for all $k \in \{1, \ldots, d\}$. We first make an assumption.

**Assumption B.1.** Any agreement is better than no agreement by a positive constant $\kappa$, i.e.,

$$u_i^\top \mathbf{x} - d_i \geq \kappa > 0 \ \forall \ i \in \{1, \ldots, n\}, \mathbf{x} \in \Delta^{d-1}. \tag{4}$$

**Lemma B.2.** *Given Assumption B.1, the negative log-Nash product, $f(\mathbf{x}) = -\sum_i \log(u_i^\top \mathbf{x} - d_i)$, is convex with respect to the joint distribution $\mathbf{x}$.*

*Proof.* We prove $f(\mathbf{x})$ is convex by showing its Hessian is positive semi-definite. First, we derive the gradient:

$$\nabla f(\mathbf{x}) = -\sum_i \frac{u_i}{u_i^\top \mathbf{x} - d_i}. \tag{5}$$

We then derive the Jacobian of the gradient to compute the Hessian. The $kl$-th entry of the Hessian is

$$H_{kl} = \sum_i \frac{u_{ik} u_{il}}{(u_i^\top \mathbf{x} - d_i)^2}. \tag{6}$$

We can write the full Hessian succinctly as

$$H = \sum_i \frac{u_i u_i^\top}{(u_i^\top \mathbf{x} - d_i)^2} = \sum_i \frac{u_i u_i^\top}{\gamma_i^2}. \tag{7}$$

Each outer product $u_i u_i^\top$ is positive semi-definite with eigenvalues $||u_i||$ (with multiplicity 1) and 0 (with multiplicity $m^n - 1$).

Each $\gamma_i$ is positive by Assumption B.1, therefore, $H$, which is the weighted sum of $u_i u_i^\top$ is positive semi-definite as well. $\square$

We also know the following.

**Lemma B.3.** *Given Assumption B.1, the infinity norm of the gradients of the negative log-Nash product are bounded by $\frac{u^{\max} n}{\kappa}$ where $u^{\max} = \max_{i,k}[u_{ik}]$.*

*Proof.* As derived in Lemma B.2, the gradient of $f(\mathbf{x})$ is

$$\nabla f(\mathbf{x}) = -\sum_i \frac{u_i}{u_i^\top \mathbf{x} - d_i}. \tag{8}$$

Using triangle inequality, we can upper bound the infinity (max) norm of the gradient as

$$||\nabla f(\mathbf{x})||_\infty = ||\sum_i \frac{u_i}{u_i^\top \mathbf{x} - d_i}||_\infty \tag{9}$$

$$\leq \sum_i ||\frac{u_i}{u_i^\top \mathbf{x} - d_i}||_\infty \tag{10}$$

$$= \sum_i \frac{||u_i||_\infty}{\gamma_i} \tag{11}$$

$$\leq \frac{u^{\max} n}{\kappa} \tag{12}$$

where $u^{\max} = \max_{i,k}[u_{ik}]$. $\square$

**Theorem B.4.** *Let $\{\mathbf{x}^t\}$ be the sequence generated by Algorithm 4 with starting point $\mathbf{x}^0 = d^{-1}\mathbf{1}$ and step size sequence $\alpha^t = \frac{\sqrt{(d-1)/d}}{||g'(\mathbf{x}^s)||_2}(t+1)^{-1/2}$. Then, for all $t > 0$ one has*

$$\max_{\mathbf{x} \in \Delta^{d-1}} g(\mathbf{x}) - \max_{0 \leq s \leq t} g(\mathbf{x}^s) \leq \frac{\sqrt{2B_\psi(\mathbf{x}^*, \mathbf{x}^0)}||g'(\mathbf{x}^s)||_2}{\sqrt{t+1}} \tag{13}$$

$$\leq \frac{\sqrt{\frac{d-1}{d}}||g'(\mathbf{x}^s)||_2}{\sqrt{t+1}} \tag{14}$$

$$\leq \frac{\sqrt{d}||g'(\mathbf{x}^s)||_\infty}{\sqrt{t+1}}. \tag{15}$$

*where $g(x) = -f(x)$ is the log-Nash product defined in Sec 4.2.*

*Proof.* Given Assumption B.1, $f(x)$ is convex (Lemma B.2) and its gradients are bounded in norm (Lemma B.3). We then apply Theorem 4.2 of (Beck & Teboulle, 2003) with $f(x) = -g(x)$, $\psi(\mathbf{x}) = \frac{1}{2}||\mathbf{x}||_2^2$ and note that $B_\psi(\mathbf{x}^*, \mathbf{x}^0) \leq \frac{1}{2}(d-1)/d$ to achieve the desired result. $\square$

**Theorem B.5.** *Let $\{\mathbf{x}^t\}$ be the sequence generated by EMDA (Beck & Teboulle, 2003) with starting point $\mathbf{x}^0 = d^{-1}\mathbf{1}$ and step size sequence $\alpha^t = \frac{\sqrt{2\log(d)}}{||g'(\mathbf{x}^s)||_\infty}(t+1)^{-1/2}$. Then, for all $t > 0$ one has*

$$\max_{\mathbf{x} \in \mathcal{X}} g(\mathbf{x}) - \max_{0 \leq s \leq t} g(\mathbf{x}^s) \leq \frac{\sqrt{2\log d}||g'(\mathbf{x}^s)||_\infty}{\sqrt{t+1}} \tag{16}$$

*where $g(x) = -f(x)$ is the log-Nash product defined in Sec 4.2.*

*Proof.* Given Assumption B.1, $f(x)$ is convex (Lemma B.2) and its gradients are bounded in norm (Lemma B.3). We then apply Theorem 5.1 of (Beck & Teboulle, 2003) with $f(x) = -g(x)$. $\square$

| Game | $n$ | Type | Description From |
|------|-----|------|------------------|
| Kuhn poker (2P) | 2 | 2P Zero-sum | (Lockhart et al., 2019) |
| Leduc poker (2P) | 2 | 2P Zero-sum | (Lockhart et al., 2019) |
| Liar's dice | 2 | 2P Zero-sum | (Lockhart et al., 2019) |
| Kuhn poker (3P) | 3 | $n$P Zero-sum | (Lockhart et al., 2019) |
| Kuhn poker (4P) | 4 | $n$P Zero-sum | (Lockhart et al., 2019) |
| Leduc poker (3P) | 3 | $n$P Zero-sum | (Lockhart et al., 2019) |
| Trade Comm | 2 | Common-payoff | (Sokota et al., 2021; Lanctot et al., 2019) |
| Tiny Bridge (2P) | 2 | Common-payoff | (Sokota et al., 2021; Lanctot et al., 2019) |
| Battleship | 2 | General-sum | (Farina et al., 2019) |
| Goofspiel (2P) | 2 | General-sum | (Lockhart et al., 2019) |
| Goofspiel (3P) | 3 | General-sum | (Lockhart et al., 2019) |
| Sheriff | 2 | General-Sum | (Farina et al., 2019) |

Table 4: Benchmark games. $n$ is the number of players. All games are available in OpenSpiel (Lanctot et al., 2019).

This argument concerns the regret of the joint version of the NBS where $\mathbf{x}$ is a correlation device. However, it also makes sense to try compute a Nash bargaining solution where players use independent strategy profiles (without any possibility of correlation across players), $\sigma = (\sigma_1 \times \cdots \sigma_n)$. In this case, $\mathbf{x}$ represents a concatenation of the individual strategies $\sigma_i$ and the projection back to the simplex is applied to each player separately. Ensuring convergence is trickier in this case, because the expected utility may not be a linear function of the parameters which may mean the function is nonconcave.

## C  Game Domain Descriptions and Details

This section contains descriptions of the benchmark games and a size estimate for Deal-or-no-Deal.

### C.1  Benchmark Games

In this section, we describe the benchmark games used in this paper. The game list is show in Table 4. As they have been used in several previous works, and are openly available, we simply copy the descriptions here citing the sources in the table.

### C.1.1  Kuhn Poker

Kuhn poker is a simplified poker game first proposed by Harold Kuhn. Each player antes a single chip, and gets a single private card from a totally-ordered $(n + 1)$-card deck, e.g. J, Q, K for the $(n = 2)$ two-player case. There is a single betting round limited to one raise of 1 chip, and two actions: pass (check/fold) or bet (raise/call). If a player folds, they lose their commitment (2 if the player made a bet, otherwise 1). If no player folds, the player with the higher card wins the pot. The utility for each player is defined as the number of chips after playing minus the number of chips before playing.

### C.1.2  Leduc Poker

Leduc poker is significantly larger game with two rounds and a two suits with $(n + 1)$ cards each, e.g. JS,QS,KS, JH,QH,KH in the $(n = 2)$ two-player case. Like Kuhn, each player initially antes a single chip to play and obtains a single private card and there are three actions: fold, call, raise. There is a fixed bet amount of 2 chips in the first round and 4 chips in the second round, and a limit of two raises per round. After the first round, a single public card is revealed. A pair is the best hand, otherwise hands are ordered by their high card (suit is irrelevant). Utilities are defined similarly to Kuhn poker.

### C.1.3   Liar's Dice

Liar's Dice(1,1) is dice game where each player gets a single private die in $\{1, \cdots, 6\}$, rolled at the beginning of the game. The players then take turns bidding on the outcomes of both dice, i.e. with bids of the form q-f referring to quantity and face, or calling "Liar". The bids represent a claim that there are at least q dice with face value f among both players. The highest die value, 6, counts as a wild card matching any value. Calling "Liar" ends the game, then both players reveal their dice. If the last bid is not satisfied, then the player who called "Liar" wins. Otherwise, the other player wins. The winner receives +1 and loser -1.

### C.1.4   Trade Comm

Trade Comm is a common-payoff game about communication and trading of a hidden item. It proceeds as follows.

1. Each player is independently dealt one of num items with uniform chance.

2. Player 1 makes one of num utterances utterances, which is observed by player 2.

3. Player 2 makes one of num utterances utterances, which is observed by player 1.

4. Both players privately request one of the num items × num items possible trades.

The trade is successful if and only if both player 1 asks to trade its item for player 2's item and player 2 asks to trade its item for player 1's item. Both players receive a reward of 1 if the trade is successful and 0 otherwise. We use num items = num utterances = 10.

### C.1.5   Tiny Bridge

A very small version of bridge, with 8 cards in total, created by Edward Lockhart, inspired by a research project at University of Alberta by Michael Bowling, Kate Davison, and Nathan Sturtevant.

This smaller game has two suits (hearts and spades), each with four cards (Jack, Queen, King, Ace). Each of the four players gets two cards each.

The game comprises a bidding phase, in which the players bid for the right to choose the trump suit (or for there not to be a trump suit), and perhaps also to bid a 'slam' contract which scores bonus points.

The play phase is not very interesting with only two tricks being played, so it is replaced with a perfect-information result, which is computed using minimax on a two-player perfect-information game representing the play phase.

The game comes in two varieties - the full four-player version, and a simplified two-player version in which one partnership does not make any bids in the auction phase.

Scoring is as follows, for the declaring partnership:

- +10 for making 1H/S/NT (+10 extra if overtrick)

- +30 for making 2H/S

- +35 for making 2NT

- -20 per undertrick

Doubling (only in the 4p game) multiplies all scores by 2. Redoubling by a further factor of 2.

An abstracted version of the game is supported, where the 28 possible hands are grouped into 12 buckets, using the following abstractions:

- When holding only one card in a suit, we consider J/Q/K equivalent

- We consider KQ and KJ in a single suit equivalent

- We consider AK and AQ in a single suit equivalent (but not AJ)

### C.1.6    Battleship

Sheriff is a general-sum variant of the classic game Battleship. Each player takes turns to secretly place a set of ships S (of varying sizes and value) on separate grids of size $H \times W$. After placement, players take turns firing at their opponent—ships which have been hit at all the tiles they lie on are considered destroyed. The game continues until either one player has lost all of their ships, or each player has completed $r$ shots. At the end of the game, the payoff of each player is computed as the sum of the values of the opponent's ships that were destroyed, minus $\gamma$ times the value of ships which they lost, where $\gamma \geq 1$ is called the loss multiplier of the game.

In this paper we use $\gamma = 2, H = 2, W = 2$, and $r = 3$. The OpenSpiel game string is:

```
battleship(board_width=2,board_height=2,
ship_sizes=[1;2], ship_values=[1.0;1.0],
num_shots=3,loss_multiplier=2.0)
```

### C.1.7    Goofspiel

Goofspiel or the Game of Pure Strategy, is a bidding card game where players are trying to obtain the most points. shuffled and set face-down. Each turn, the top point card is revealed, and players simultaneously play a bid card; the point card is given to the highest bidder or discarded if the bids are equal. In this implementation, we use a fixed deck of decreasing points. In this paper, we an imperfect information variant where players are only told whether they have won or lost the bid, but not what the other player played. The utility is defined as the total value of the point cards achieved.

In this paper we use $K = 4$ card decks. So, e.g. the OpenSpiel game string for the three-player game is:

```
goofspiel(imp_info=True,returns_type=total_points,
players=3,num_cards=4)
```

### C.1.8    Sheriff

Sheriff is a simplified version of the Sheriff of Nottingham board game. The game models the interaction of two players: the Smuggler—who is trying to smuggle illegal items in their cargo– and the Sheriff– who is trying to stop the Smuggler. At the beginning of the game, the Smuggler secretly loads his cargo with $n \in \{0, ..., n_{max}\}$ illegal items. At the end of the game, the Sheriff decides whether to inspect the cargo. If the Sheriff chooses to inspect the cargo and finds illegal goods, the Smuggler must pay a fine worth $p \cdot n$ to the Sheriff. On the other hand, the Sheriff has to compensate the Smuggler with a utility if no illegal goods are found. Finally, if the Sheriff decides not to inspect the cargo, the Smuggler's utility is $v \cdot n$ whereas the Sheriff's utility is 0. The game is made interesting by two additional elements (which are also present in the board game): bribery and bargaining. After the Smuggler has loaded the cargo and before the Sheriff chooses whether or not to inspect, they engage in r rounds of bargaining. At each round $i = 1, ..., r$, the Smuggler tries to tempt the Sheriff into not inspecting the cargo by proposing a bribe $b_i \in \{0, ...b_{max}\}$, and the Sheriff responds whether or not they would accept the proposed bribe. Only the proposal and response from round r will be executed and have an impact on the final payoffs—that is, all but the $r^{th}$ round of bargaining are non-consequential and their purpose is for the two players to settle on a suitable bribe amount. If the Sheriff accepts bribe $b_r$, then the Smuggler gets $p \cdot n - b_r$, while the Sheriff gets $b_r$.

In this paper we use values of $n_{max} = 10, b_{max} = 2, p = 1, v = 5$, and $r = 2$. The OpenSpiel game string is:

```
sheriff(item_penalty=1.0,item_value=5.0,
max_bribe=2,max_items=2,num_rounds=2,
sheriff_penalty=1.0)
```

| Hyperparameter Name | Values in DoND | Values in CT (PSRO w/ DQN BR) |
|---|---|---|
| AlphaZero training episodes $num\_eps$ | $10^4$ | $10^4$ |
| Network input representation | an infoset vector $\in \mathbb{R}^{309}$ implemented in (Lanctot et al., 2019) | an infoset vector $\in \mathbb{R}^{463}$ |
| Network size for policy & value nets | [256, 256] for the shared torso | [1024, 512, 256] |
| Network optimizer | SGD | SGD |
| UCT constant $c_{uct}$ | 20 for IR and IE, 40 for SW, 100 for NP | |
| Returned policy type | greedy towards $v$ | argmax over learned q-net |
| # simulations in search | 300 | |
| Learning rate for policy & value net | 2e-3 for IR and IE, 1e-3 for SW, 5e-4 for NP | 1e-3 |
| Delayed # episodes for replacing $v, p, g$ | 200 | 200 |
| Replay buffer size $|D|$ | $2^{16}$ | |
| Batch size | 64 | 128 |
| PSRO empirical game entries # simulation | 200 | 2000 |
| L2 coefficient in evaluator $c_1$ | 1e-3 | |
| Network size for generator net | [300, 100] MLP | |
| Learning rate for generator net | 1e-3 | |
| L2 coefficient in generator net $c_2$ | 1e-3 | |
| PSRO minimum pure-strategy mass lower bound | 0.005 | 0.005 |

Table 5: Hyper-parameters.

## C.2 Approximate Size of Deal or No Deal

To estimate the size of Deal or No Deal described in Section 5.3, we first verified that there are 142 unique preference vectors per player. Then, we generated 10,000 simulations of trajectories using uniform random policy computing the average branching factor (number of legal actions per player at each state) as $b \approx 23.5$.

Since there are 142 different information states for player 1's first decision, about $142bb$ player 1's second decision, etc. leading to $142(1 + b^2 + b^4 + b^8) \approx 13.2 \times 10^{12}$ information states. Similarly, player 2 has roughly $142(1 + b^1 + b^3 + b^5 + b^7) = 5.63 \times 10^{11}$ information states.

# D  Hyper-parameters and Algorithm Settings

We provide detailed description of our algorithm in this section. Our implementation differs from the original ABR (Timbers et al., 2022) and AlphaZero (Silver et al., 2018) in a few places. First, instead of using the whole trajectory as a unit of data for training, we use per-state pair data. We maintain separate data buffers for policy net, value net, and generate net, and collect corresponding $(s_i, r)$, $(s_i, p)$, $(s_i, h)$ to them respectively. On each gradient step we will sample a batch from each of these data buffer respectively. The policy net and value net shared a torso part, while we keep a separate generative net (detailed in Section D.1). The policy net and value net are trained by minimizing loss $(r - v)^2 - \pi^{*T} \log p + c_1 \|\theta\|_2^2$, where (1) $\pi^*$ are the policy targets output by the MCTS search, (2) $p$ are the output by the policy net, (3) $v$ are the output by the value net, (4) $r$ are the outcome value for the trajectories (5) $\theta$ are the neural parameters. During selection phase the algorithm select the child that maximize the following PUCT (Silver et al., 2018) term: $\texttt{MaxPUCT}(s, \boldsymbol{p}) = \arg\max_{a \in \mathcal{A}(s)} \frac{s.child(a).value}{s.child(a).visits} + c_{uct_c} \cdot \boldsymbol{p}(s)_a \cdot \frac{\sqrt{s.total\_visits}}{s.child(a).visits+1}$. Other hyperparameters are listed in Table 5.

## D.1  Generative Models

### D.1.1  Deal or No Deal

In DoND given an information state, the thing we only need to learn is the opponent's hidden utilities. And since the utilities are integers from 0 to 10, we design our generative model as a supervised classification task. Specifically, we use the 309-dimensional state $s_i$ as input and output three heads. Each head corresponds to one item of the game. each head consists of 11 logits corresponding to each of 11 different utility values. Then each gradient step we sample a batch of $(s_i, h)$ from the data buffer. And then do one step of gradient descent to minimize the cross-entropy loss $-h^T \log \boldsymbol{g}(s) + c_2 \|\theta'\|_2^2$, where here we overload $h$ to represent a one-hot encoding of the actual opponent utility vectors. And each time we need to generate a new state at information state $s_i$, we just feed forward $s_i$ and sample the utilities according to the output logits. But since we have a constraints $\mathbf{v}_1 \cdot \mathbf{p} = \mathbf{v}_2 \cdot \mathbf{p} = 10$, the sampled utilities may not always be feasible under this

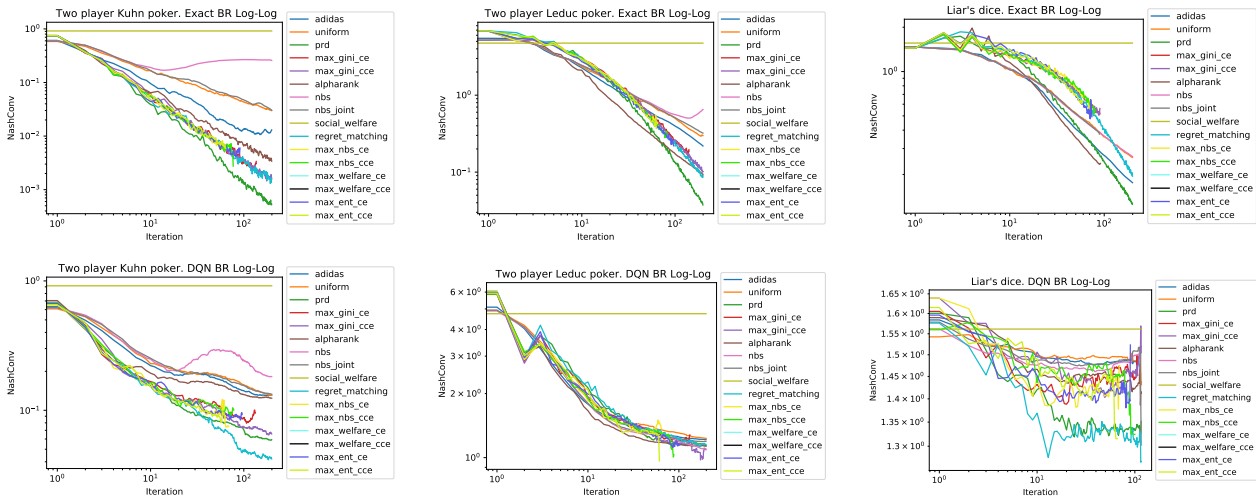

Figure 7: Empirical Convergence to Nash Equilibria using Exact vs. DQN Best Response versus in Two-Player Zero-Sum Benchmark Games.

constraint. Then we will do an additional L2 projection on to all feasible utility vector space and then get the final results.

## E   Additional Results

The full analysis of empirical convergence to approximate Nash equilibrium and social welfare are shown for various settings:

- Two-player Zero-Sum Games: Figure 7.
- $n$-player Zero-Sum Games: Figure 8.
- Common Payoff Games: Figure 9.
- General-Sum Games: Figure 10.

### E.1   Colored Trails

The full Pareto gap graphs as a function of training time is shown in Figure 11. Some examples of the evolution of expected outcomes over the course of PSRO training are shown in Figure 12.

### E.2   Deal or No Deal

In this section, we described how we train and select the PSRO agents in DoND human behavioral studies. Due the experimental limitation, we can only select 5 of our agents to human experiments. For convenience from now on we label a PSRO agent as ($MSS$, $BACKPROP\_TYPE$, $FINAL\_TYPE$) if it is trained under meta-strategy solver $MSS$, its back-propagation type is $BACKPROP\_TYPE$ and we use $FINAL\_TYPE$ as its decision architecture. We apply standard empirical game theoretic analysis (Wellman, 2006; Jordan et al., 2007) on our agent pool: we create a 113x113 symmetric empirical game by simulating head to head results between every pair of our agents. During each simulation we toss a coin to assign the roles (first-mover or second-mover in DoND) to our agents. Then we decide the selected agents based on this empirical game matrix.

Specifically, we want to select: (1) the most competitive agents (2) the most collaborative agents and (3) the fairest agent in our pool in a principled way. Now we explain how we approach these criterions one by one.

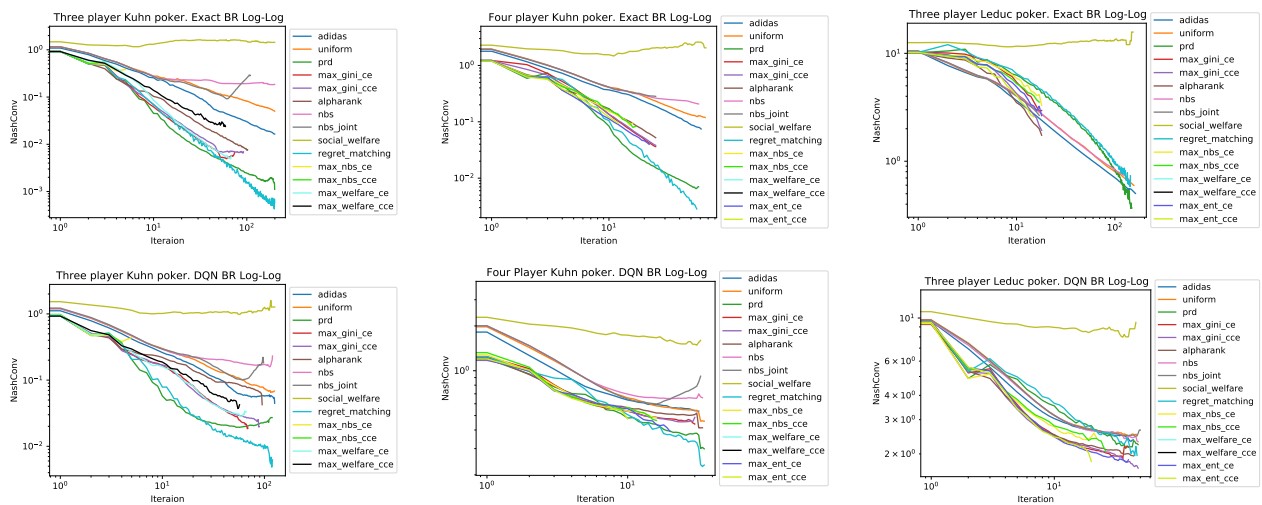

Figure 8: Empirical Convergence to Nash Equilibria using Exact vs. DQN Best Response in $n$-Player Zero-Sum Benchmark Games.

| Agent \ Opponent | IndRL | Com1 | Com2 | Coop | Fair |
|---|---|---|---|---|---|
| IndRL | 4.85 | 4.98 | 5.02 | 7.05 | 6.96 |
| Com1 | 3.75 | 4.97 | 4.66 | 7.19 | 6.70 |
| Com2 | 3.20 | 5.30 | 5.04 | 6.84 | 6.86 |
| Coop | 5.63 | 4.43 | 4.32 | 6.67 | 6.64 |
| Fair | 5.47 | 4.43 | 4.19 | 6.59 | 6.52 |

Table 6: Head-to-head empirical game matrix among our selected agents . The $(i, j)$-th entry is the payoff of the $i$-th agent when it is playing with the $j$-th agent.

For (1), we apply our competitive MSSs (e.g., ADIDAS, CE/CCE solvers) on this empirical game matrix and solve for a symmetric equilibrium. Then we rank the agents according to their expected payoff when the opponents are playing according to this equilibrium. This approach is inspired by Nash-response ranking in (Jordan et al., 2007) and Nash averaging in (Balduzzi et al., 2018) which is recently generalized to any $N$-player general-sum games (Marris et al., 2022). We found that Independent DQN, (MGCE, IA, SP) (denoted as Comp1), (MGCCE, IR, SP) (denoted as Comp2) rank at the top under almost all competitive MSS.

For (2), we create two collaborative games where the payoffs of agent1 v.s. agent2 are their social welfare/Nash product. We conduct the same analysis as in (1), and found the agent (uniform, NP, RP) normally ranks the first (thus we label it as Coop agent).

For (3), we create an empirical game where the payoffs of agent1 v.s. agent2 are the negative of their absolute payoff difference, and apply the same analysis. We also conduct a Borda voting scheme: we rank the agent pairs in increasing order of their absolute payoff difference. Each of these agent pairs will got a Borda voting score. Then the score of an agent is the summation of the scores of agents pairs that this agent is involved in. In both approaches, we find the agent (MGCE, NP, RP) ranks the top. Therefore we label it as Fair agent.

To summary, the five agents we finally decided to conduct human experiments are (1) DQN trained through self-play (IndRL) (2) (MGCE, IA, SP) (Comp1) (3) (MGCCE, IR, SP) (Comp2) (4) (uniform, NP, RP) (Coop) (5) (MGCE, NP, RP) (Fair).

We show the head-to-head empirical game between these five agents in Table 6, social-welfare in Table 7, empirical Nash product in Table 8.

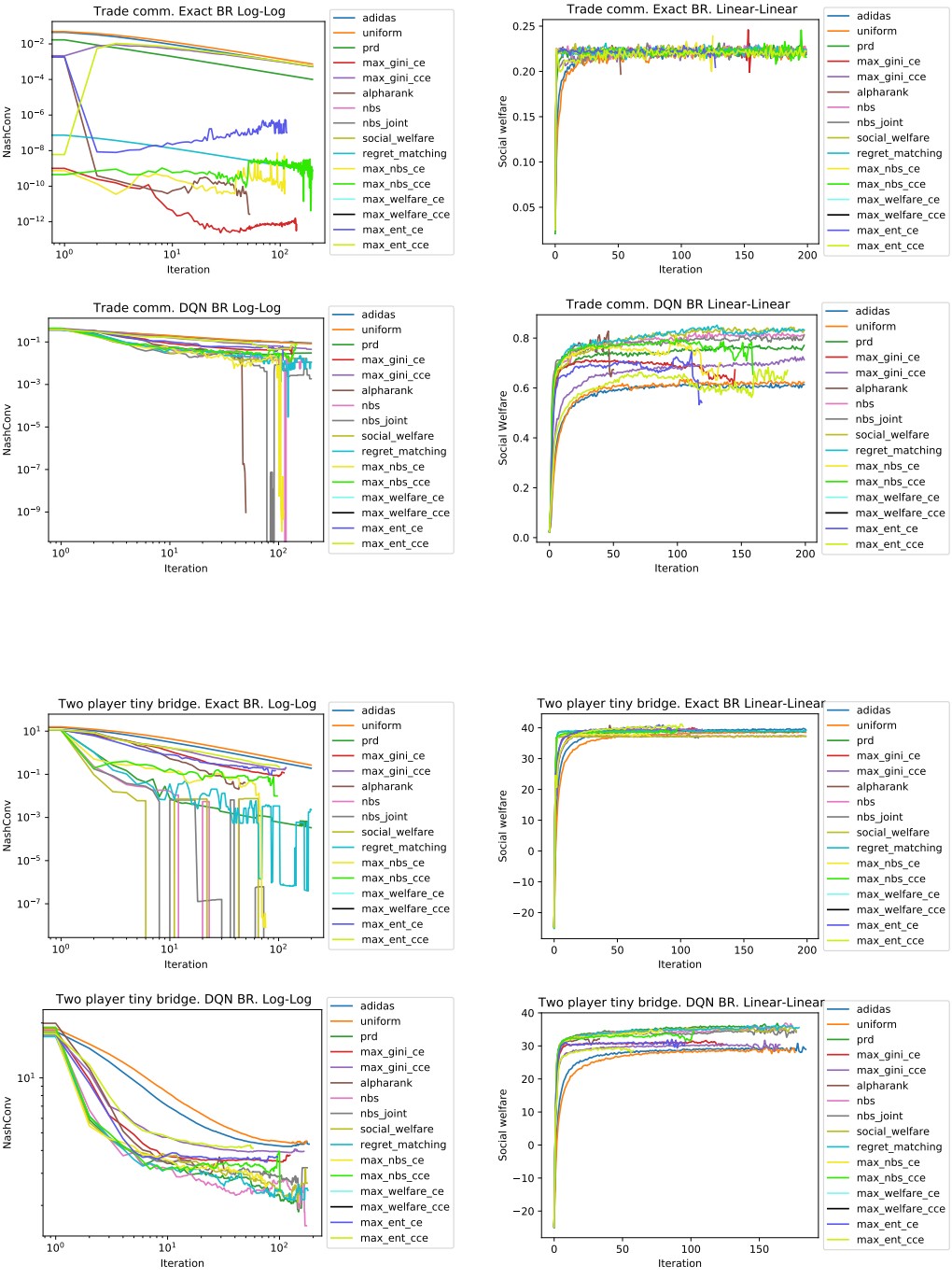

Figure 9: Empirical Convergence to Nash Equilibria and Social Welfare in Common Payoff Benchmark Games.

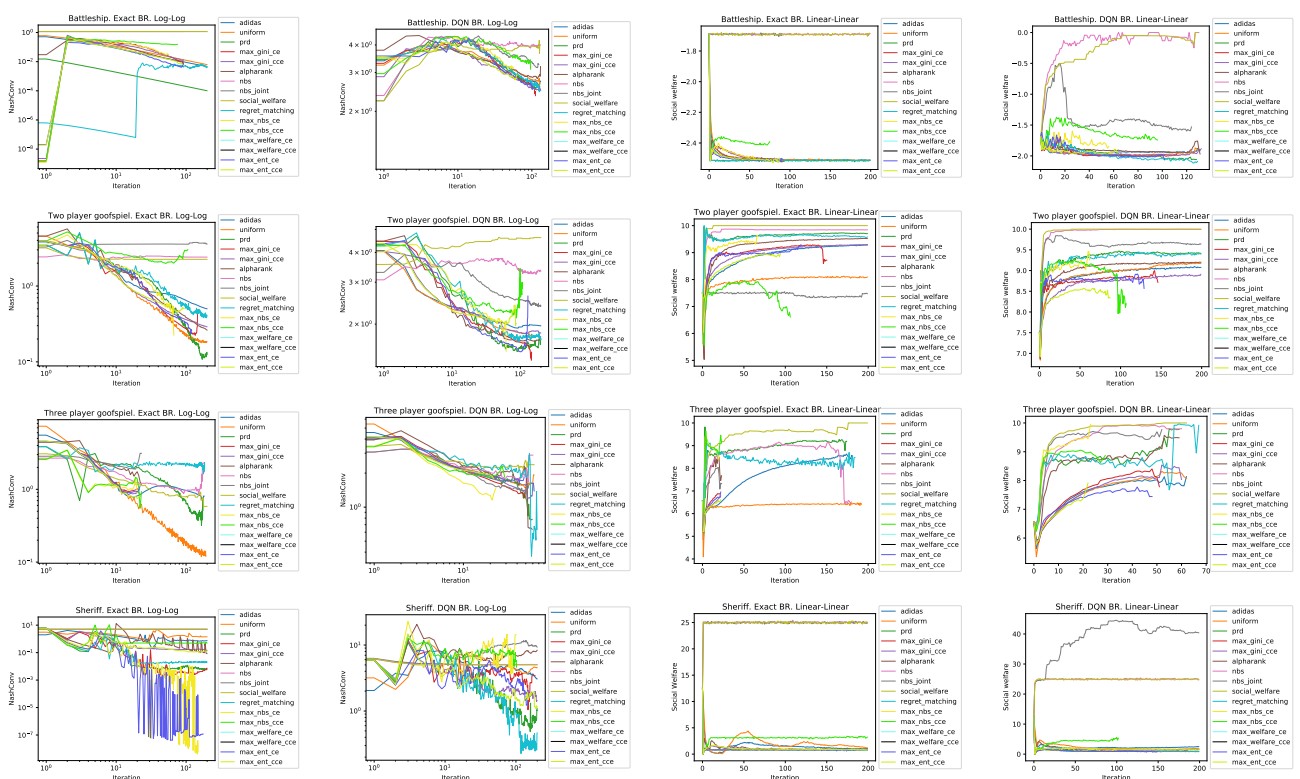

Figure 10: Empirical Convergence to Nash Equilibria and Social Welfare in General-Sum Benchmark Games.

| Agent \ Opponent | IndRL | Com1 | Com2 | Coop | Fair |
|---|---|---|---|---|---|
| IndRL | 9.70 | 8.73 | 8.22 | 12.68 | 12.42 |
| Com1 | 8.73 | 9.94 | 9.96 | 11.62 | 11.12 |
| Com2 | 8.22 | 9.96 | 10.09 | 11.16 | 11.05 |
| Coop | 12.68 | 11.62 | 11.16 | 13.34 | 13.22 |
| Fair | 12.42 | 11.12 | 11.05 | 13.22 | 13.05 |

Table 7: Head-to-head empirical social welfare matrix among our selected agents . The $(i, j)$-th entry is the social welfare of when $i$-th agent is playing with the $j$-th agent.

# F   Human Behavioral Studies

## F.1   Study Design

The protocol for the human behavioral studies underwent independent ethical review and was approved by the institutional review board at our institution. All participants provided informed consent before joining the study.

We collected data through "tournaments" of Deal or No Deal games in two different conditions: Human vs. Human (HvH) and Human vs. Agent (HvA). Upon joining the study, participants were randomly assigned to one of the two conditions. Participants in both conditions proceeded through the following sequence of steps:

1. Read study instructions and gameplay tutorial (Figures 14–18).

2. Take comprehension test (Figures 19 & 20).

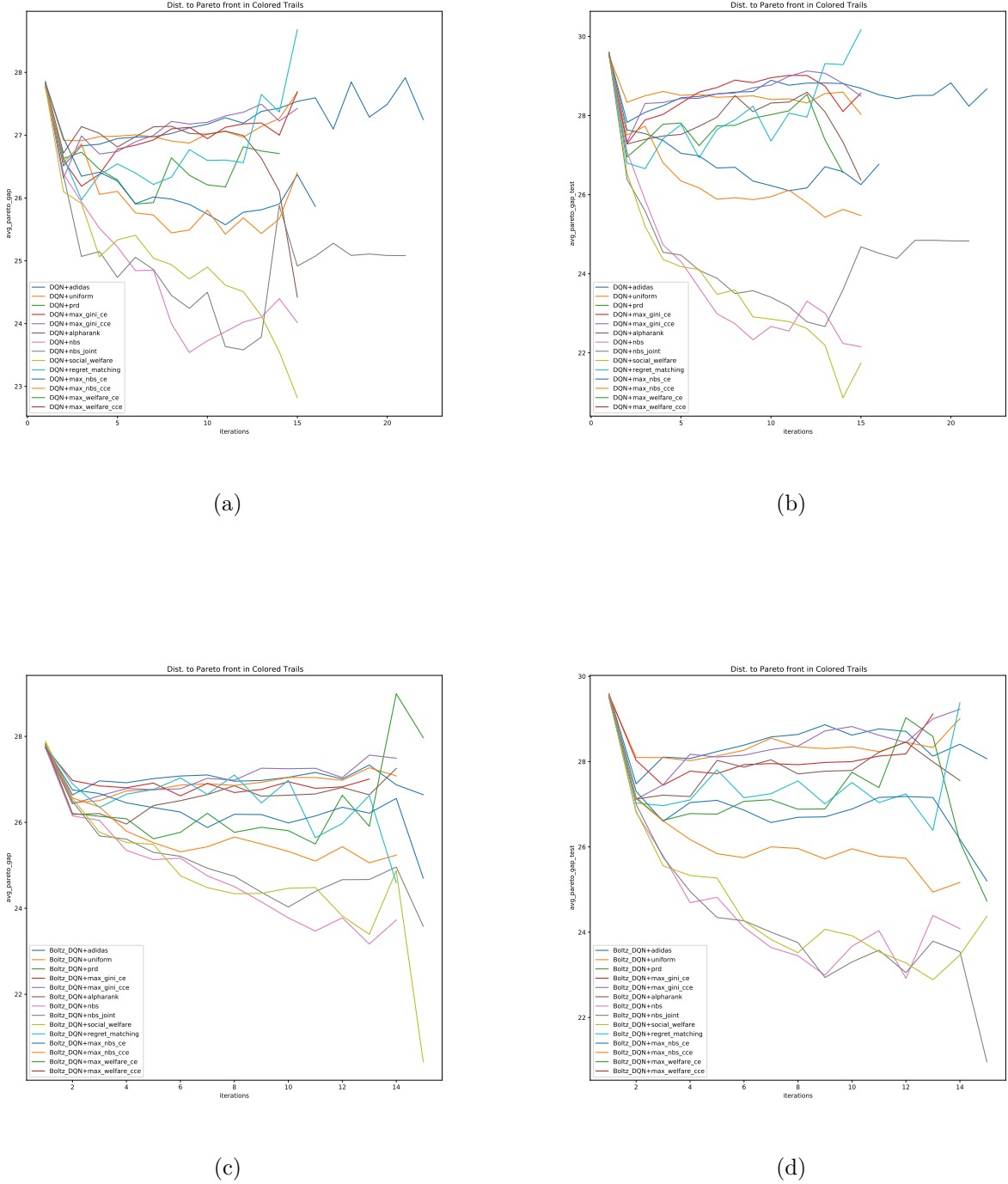

Figure 11: Average Pareto gap using DQN best response (top: (a) and (b)) and Boltzmann DQN (bottom: (c) and (d)), training gap (left: (a) and (c)) and gap on held-out test boards (right: (b) and (d)).

3. Wait for random assignment to a tournament with five other participants (HvH) or wait for agents to load for a tournament (HvA; Figure 21).

4. Play episode of Deal or No Deal game (Figure 22).

| Agent \ Opponent | IndRL | Com1 | Com2 | Coop | Fair |
|---|---|---|---|---|---|
| IndRL | 23.48 | 18.69 | 16.05 | 39.66 | 38.02 |
| Com1 | 18.69 | 24.70 | 24.68 | 31.84 | 29.63 |
| Com2 | 16.05 | 24.68 | 25.44 | 29.57 | 28.73 |
| Coop | 39.66 | 31.84 | 29.57 | 44.51 | 43.70 |
| Fair | 38.02 | 29.63 | 28.74 | 43.70 | 42.56 |

Table 8: Head-to-head empirical Nash product matrix among our selected agents . The $(i, j)$-th entry is the Nash product when $i$-th agent is playing with the $j$-th agent.

5. See score confirmation for last episode and wait for next episode (Figure 23a).

6. Repeat steps 4 and 5 for four additional episodes.

7. Note total earnings and transition to post-game questionnaire (Figure 23b).

We required participants to answer all four questions in the comprehension test correctly to continue to the rest of the study. The majority of participants (71.4%) passed the test and were randomly sorted into tournaments in groups of $n = 6$ (for the HvH condition) or $n = 1$ (for the HvA condition). We provided the remainder a show-up payment for the time they spent on the study tutorial and test.

The participants played DoND for real monetary stakes, receiving an additional $0.10 of bonus for each point they earned in the study. To ensure that one non-responsive participant did not disrupt the study for any other participants in their tournament, episodes had a strict time limit of 120 seconds. After the time limit for a given episode elapsed, all uncompleted games were marked as "timed out" and any participants in the tournament were moved to the next step of the study.

We apply two exclusion criteria to build our final datasets of game episodes: first, exclude all episodes that timed out before reaching a deal or exhausting all turns; and second, exclude all episodes involving a participant who was non-responsive during the tournament (i.e., did not take a single action in a episode).

In the HvH condition, 228 participants passed the comprehension test and were sorted into tournaments. Eleven participants were non-responsive and an additional 6.3% of games timed out before reaching a deal or exhausting all turns, resulting in a final sample of $N = 217$ participants (39.5% female, 59.1% male, 0.9% trans or nonbinary; median age range: 30–40).

In HvH tournaments, each participant plays one episode against each of the five other participants in their group. We use a round-robin structure to efficiently match participants for each episode, ensuring each game involves two participants who have not interacted before. We randomize player order in each game. The final HvH dataset contains $k = 483$ games.

In the HvA condition, 130 participants passed the comprehension test and were sorted into tournaments. One participant was non-responsive and an additional 15.2% of games timed out before reaching a deal or exhausting all turns, resulting in a final sample of $N = 129$ participants (44.5% female, 53.1% male, 0.8% trans or nonbinary; median age range: 30–40).

In HvA tournaments, each participant plays one episode against each of the five agents evaluated in the study. We randomize agent order in each tournament and player order in each game. The final HvA dataset contains $k = 547$ games.

During each episode of the HvA tournaments, agent players waited a random amount of time after participant actions to send their own actions. The agents randomly sampled their time delays from a normal distribution with a mean of 10 seconds and a standard deviation of 3 seconds.

After completing all episodes in their tournament, participants proceeded to complete a post-task questionnaire that included the slider measure of Social Value Orientation (Murphy et al., 2011), demographic questions, and open-ended questions soliciting feedback on the study.

Participants completed the study in an average of 18.2 minutes. The base pay for the study was \$2.50, with an average performance bonus of \$3.70.

### F.2 Analysis

We fit linear mixed- and random-effects models to estimate and compare the average returns generated by our agents and by study participants.

We first fit a linear mixed-effects model using just the HvA data, predicting human return from each episode with one fixed effect (a categorical variable representing each type of agent playing in the episode) and one random effect (representing each participant in the HvA condition). The effect estimates (the mean points earned by a human player against each agent) are shown in the $\bar{u}_{\text{Humans}}$ column of Table 3. We estimate 95% confidence intervals for the individual effects through bootstrapping with 500 resamples.

We next fit a linear mixed-effects model using just the HvA data, predicting agent return from each episode with one fixed effect (a categorical variable representing each type of agent playing in the episode) and one random effect (representing each participant in the HvA condition). The effect estimates (the mean points earned by each agent) are shown in the $\bar{u}_{\text{Agent}}$ column of Table 3. We estimate 95% confidence intervals for the individual effects through bootstrapping with 500 resamples.

We similarly fit a linear mixed-effects model using just the HvA data, predicting social welfare (average return) from each episode with one fixed effect (a categorical variable representing each type of agent playing in the episode) and one random effect (representing each participant in the HvA condition). The effect estimates (the mean social welfare catalyzed by each agent) are shown in the $\bar{u}_{\text{Comb}}$ column of Table 3. We estimate 95% confidence intervals for the individual effects through bootstrapping with 500 resamples.

We then fit a linear random-effects model using just the HvH data, predicting the return for one player (randomly selected) in each episode with one random effect (representing each participant in the HvH condition). Participants in the HvH condition achieve an individual return of 6.93 [6.72, 7.14], on expectation. We estimate the 95% confidence interval for this effect through bootstrapping with 500 resamples.

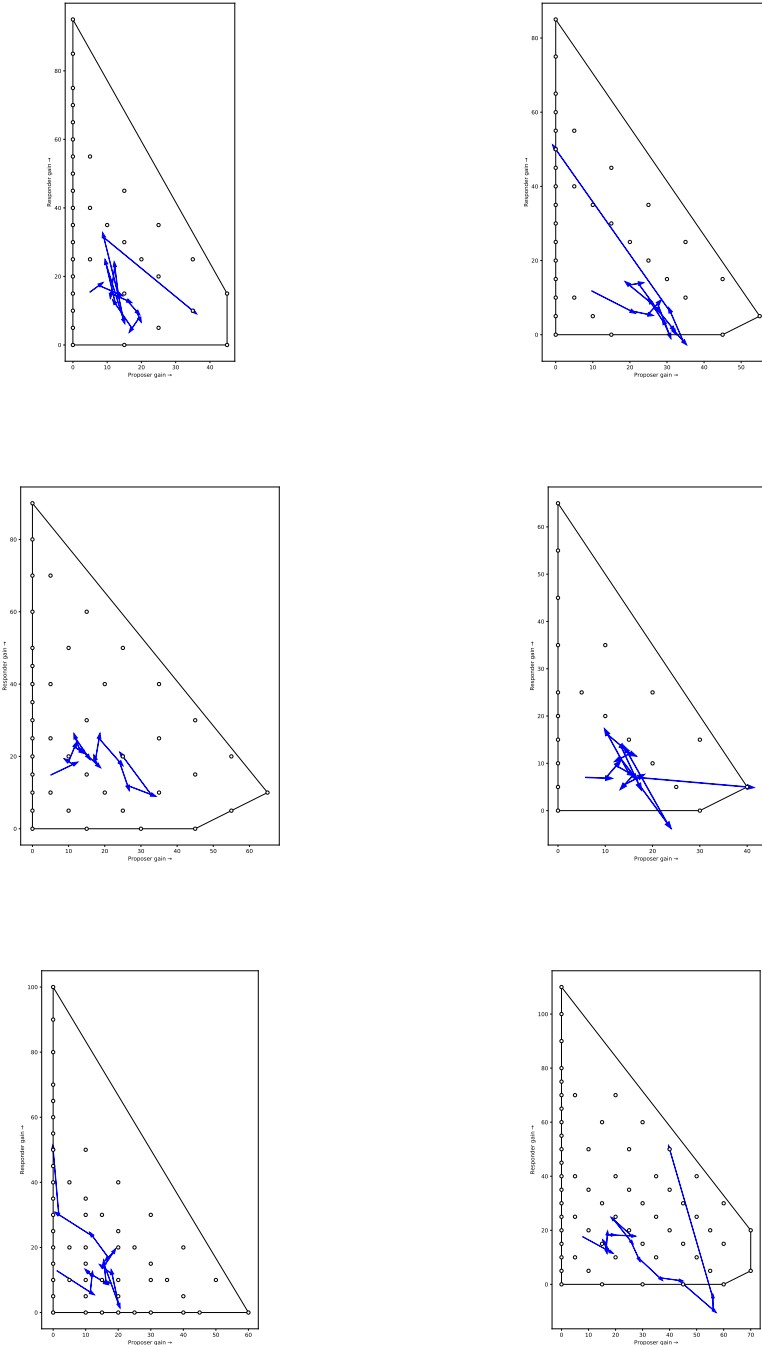

Figure 12: Evolution of the expected outcomes of the PSRO agents using the DQN best response type and social welfare MSS. Each diagram depicts the outcome of the agent for a single configuration of Colored Trails: circles represent the rational outcomes (pure joint strategies where players have non-negative gain). The outer surface of the convex hull represents the Pareto front/envelope. To make a 2D image, the proposers' gains are aggregated and only the winning proposer's value is included in the outcome computation. The blue directed path represents the PSRO agents' expected outcomes at iterations $t \in \{0, 1, \cdots, 15\}$, where each point estimated from 100 samples. Note that values can be negative due to sampling approximation but also due to choosing legal actions that result in negative gain.

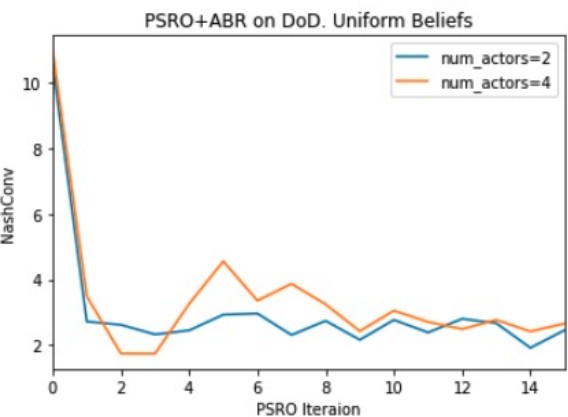

Figure 13: Value of exploits found by ABR with uniform beliefs against search-enhanced PSRO agents. The y-axis here is an approximation of NashConv in the same sense as used in (Timbers et al., 2022).

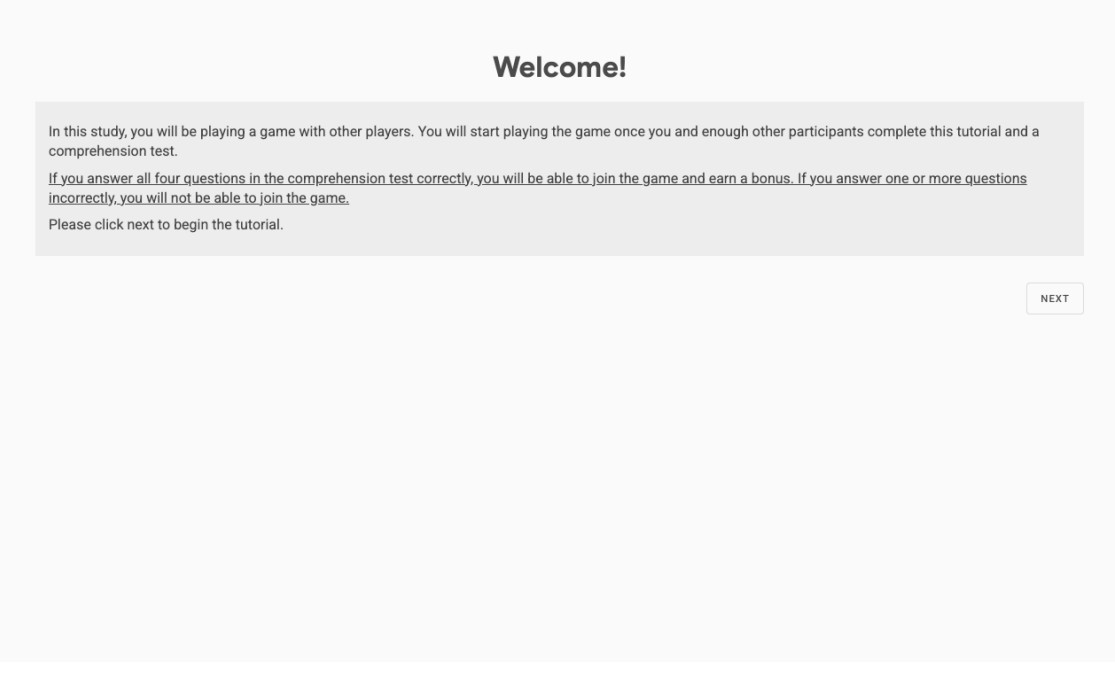

(a) Screen 1: Welcome participants to the experiment.

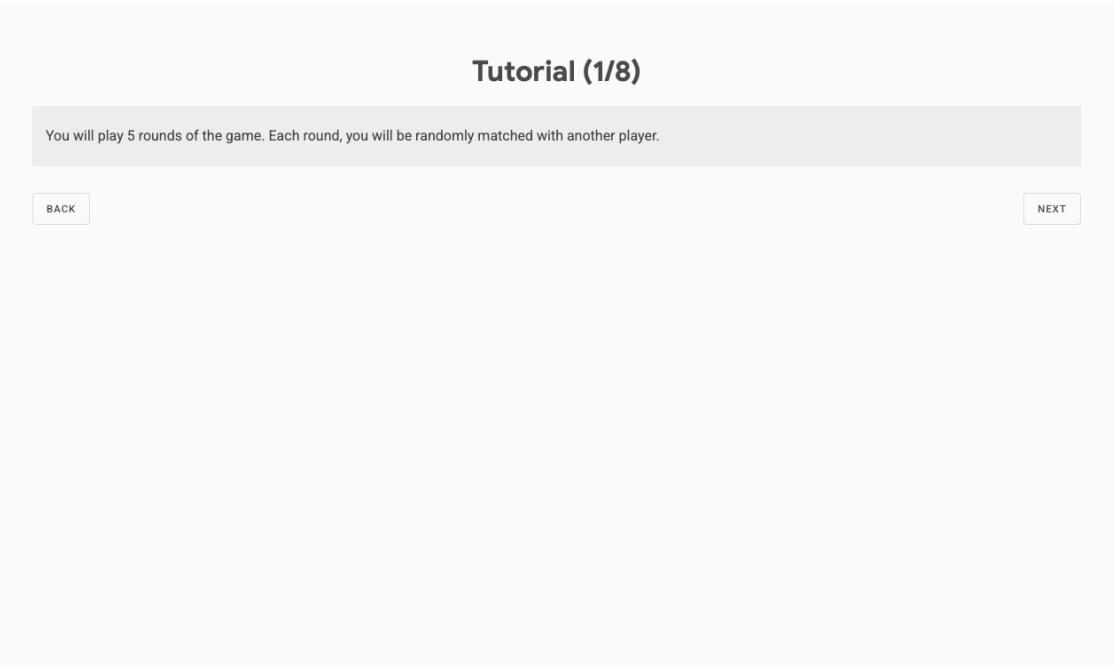

(b) Screen 2: Explain study length and player matching.

Figure 14: Screenshots of the participant interface for the "Deal or No Deal" study. (a) The participant reads general study information. (b) The participant reads tutorial information about the game.

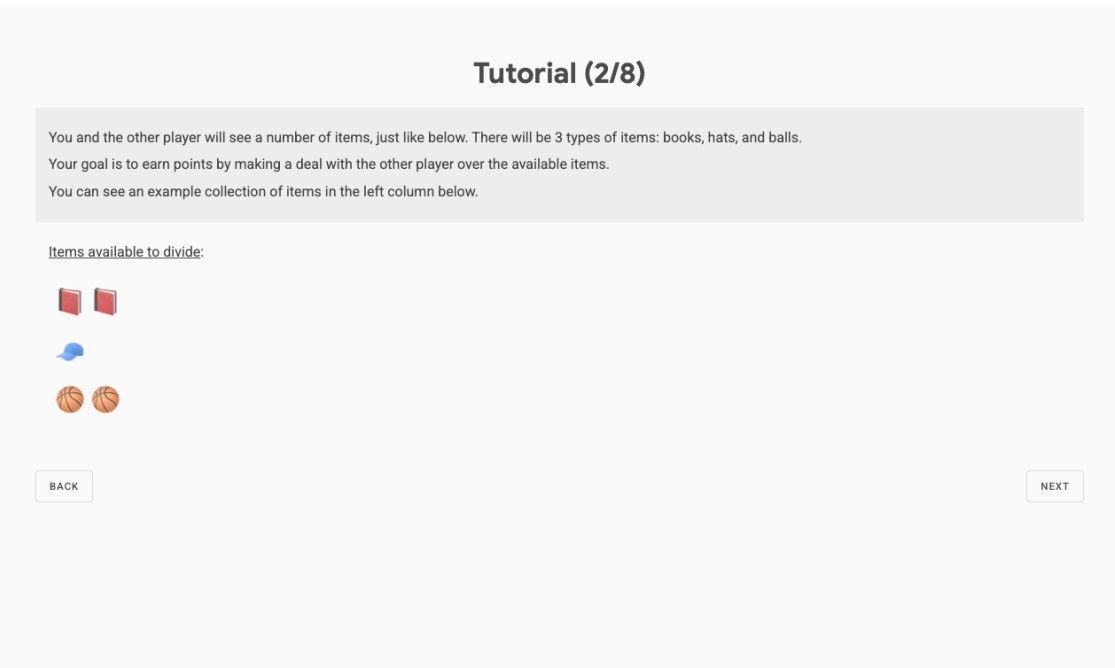

(a) Screen 3: Explain items.

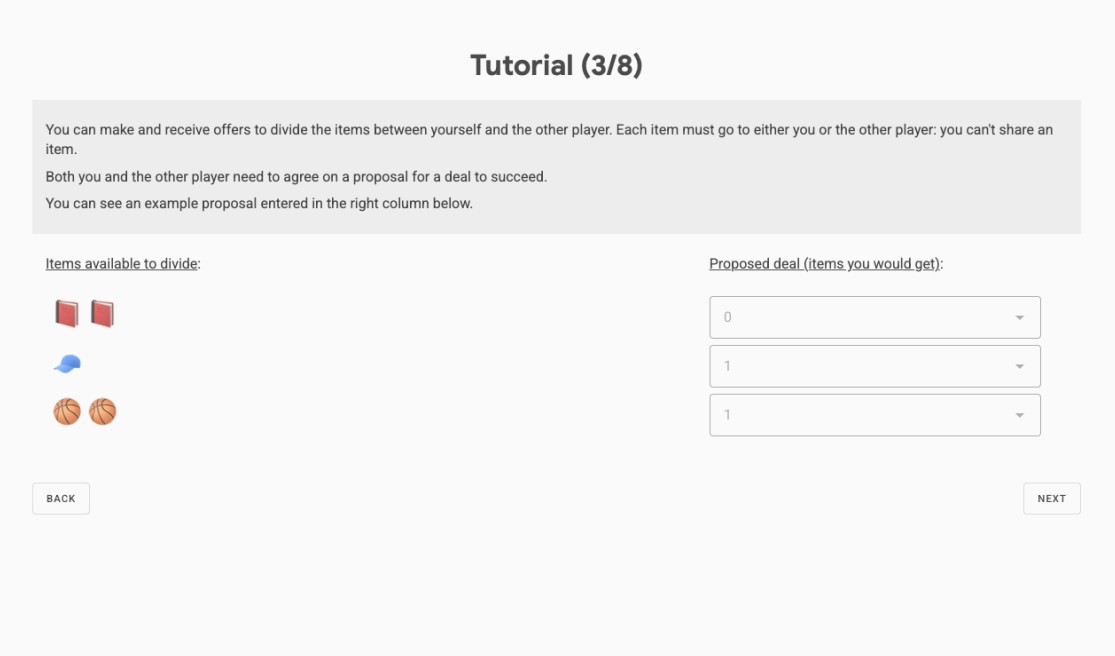

(b) Screen 4: Explain the goal of item distribution.

Figure 15: Screenshots of the participant interface for the "Deal or No Deal" study. (a) The participant reads tutorial information about the game. (b) The participant reads tutorial information about the game.

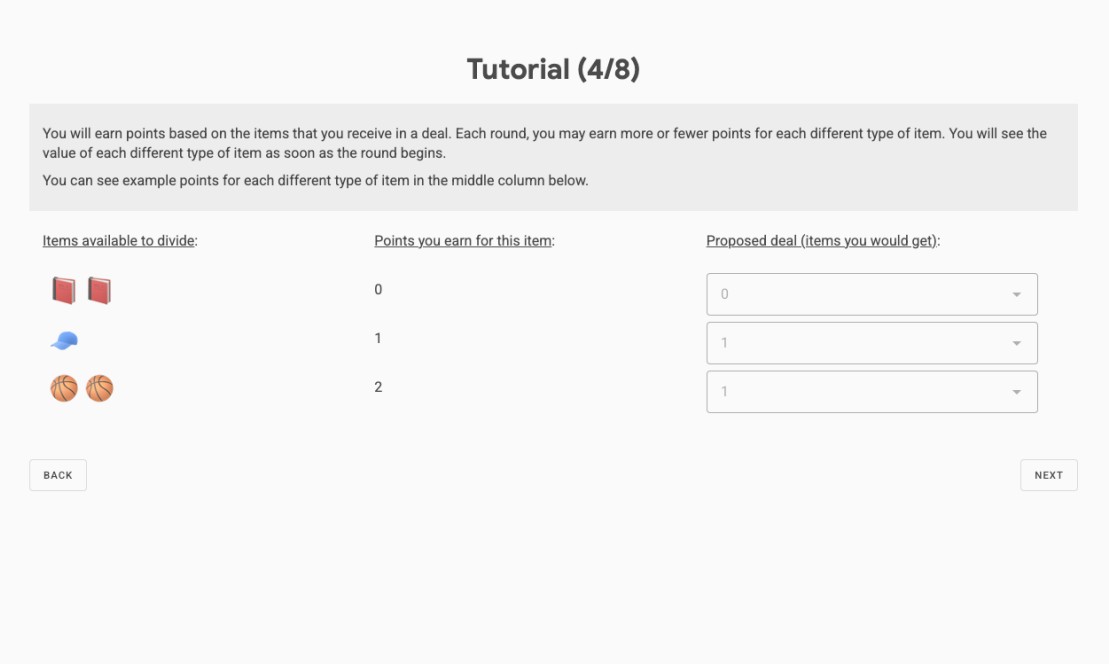

(a) Screen 5: Explain points.

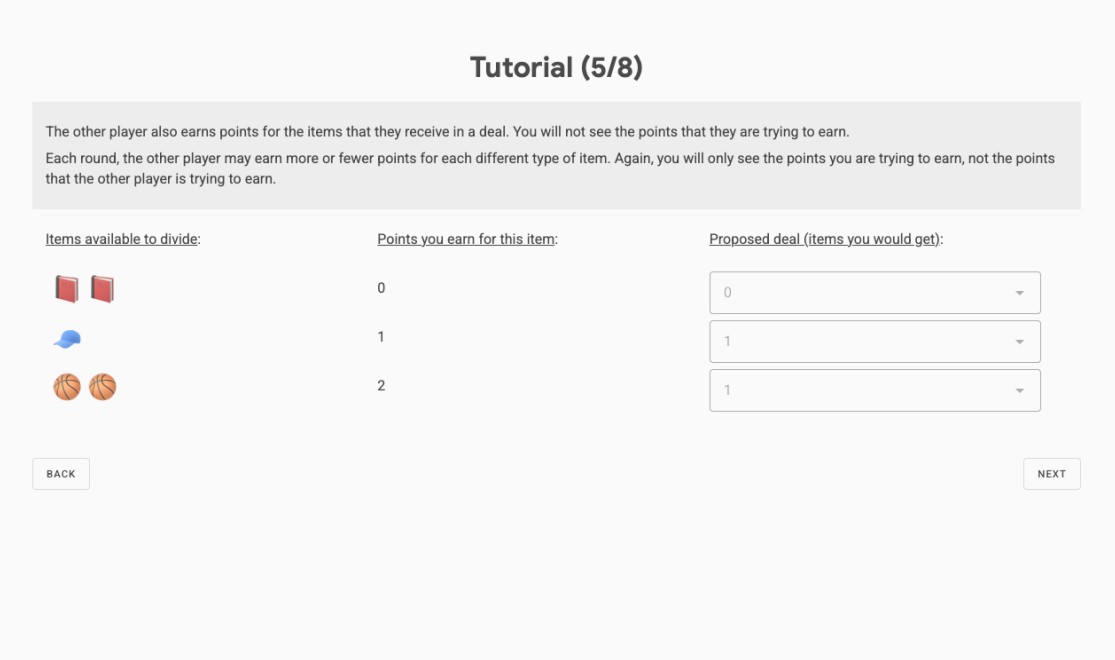

(b) Screen 6: Explain other player's points.

Figure 16: Screenshots of the participant interface for the "Deal or No Deal" study. (a) The participant reads tutorial information about the game. (b) The participant reads tutorial information about the game.

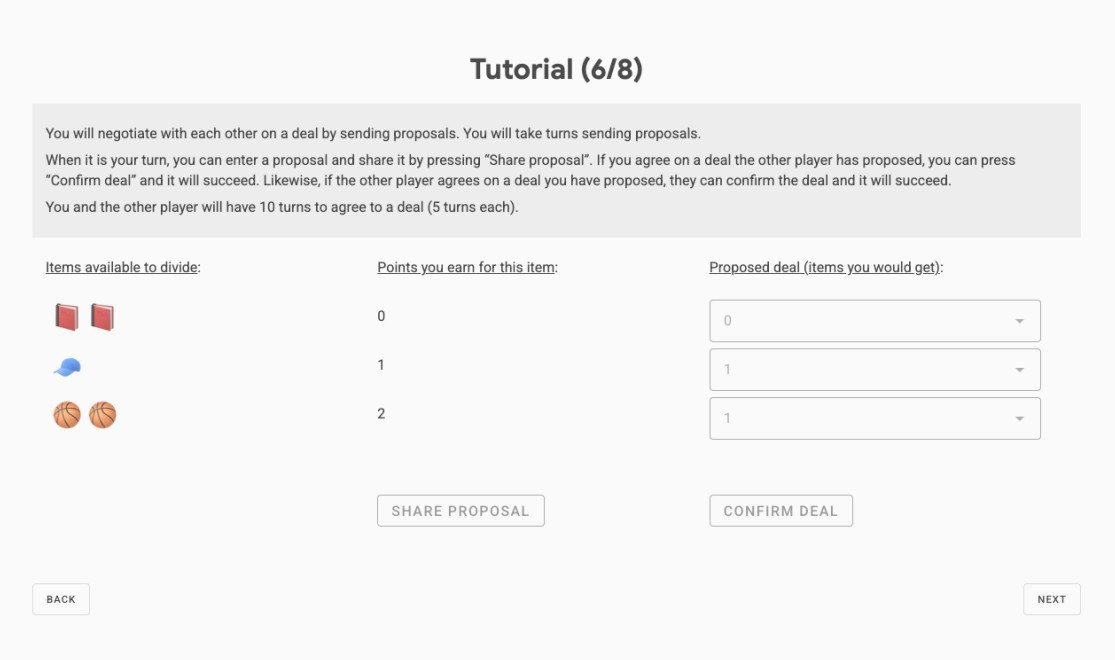

(a) Screen 7: Explain actions.

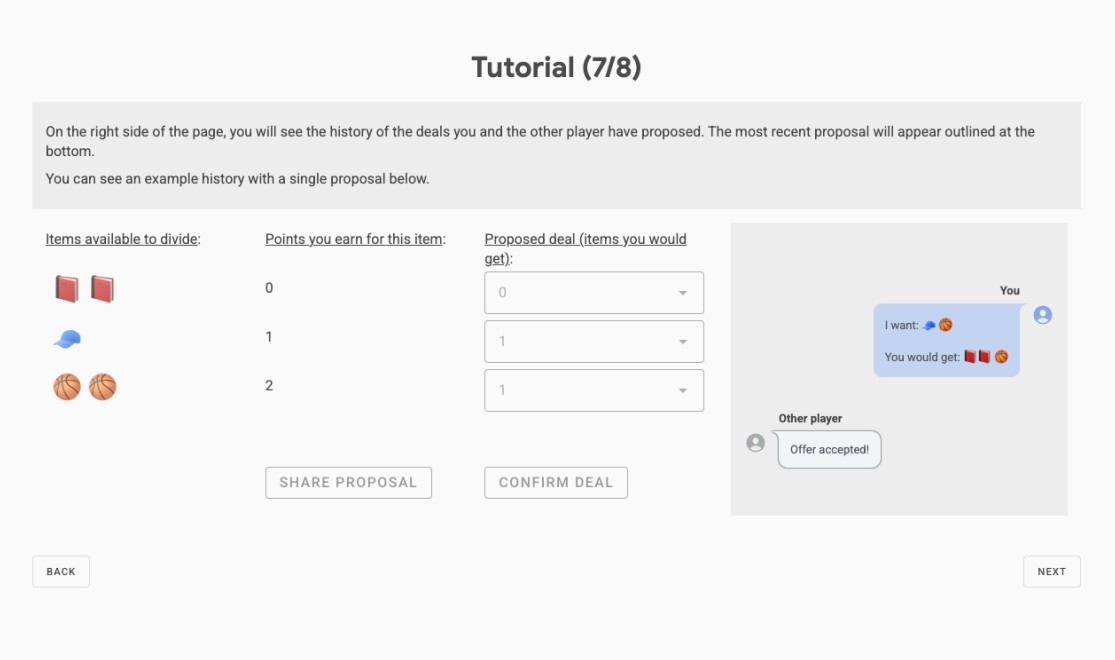

(b) Screen 8: Explain communication.

Figure 17: Screenshots of the participant interface for the "Deal or No Deal" study. (a) The participant reads tutorial information about the game. (b) The participant reads tutorial information about the game.

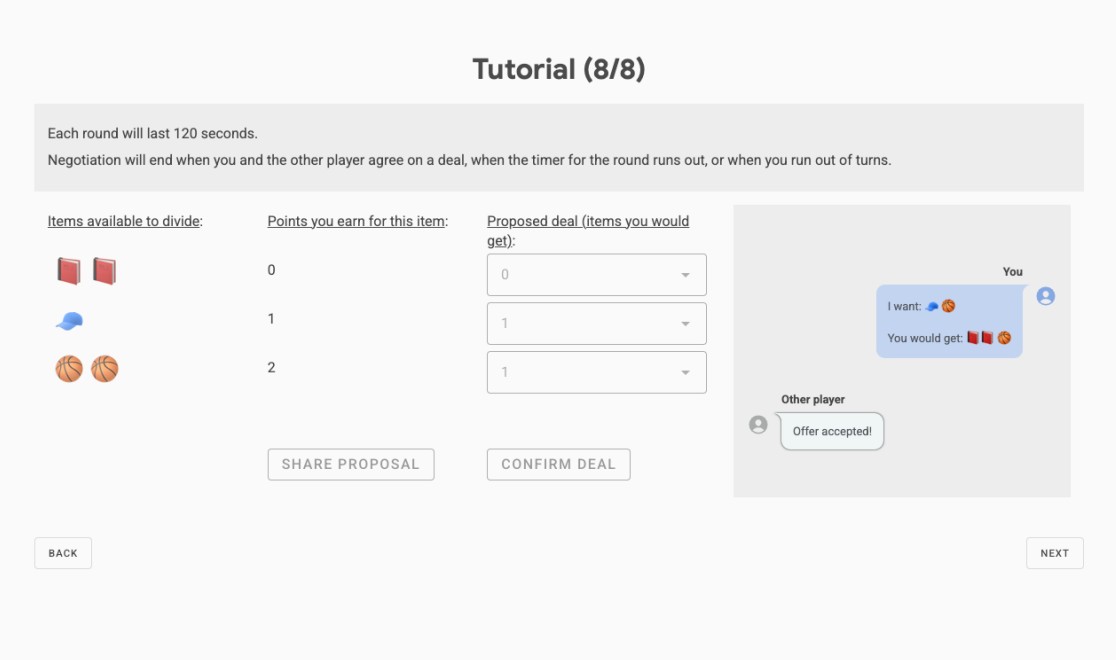

(a) Screen 9: Explain episode length.

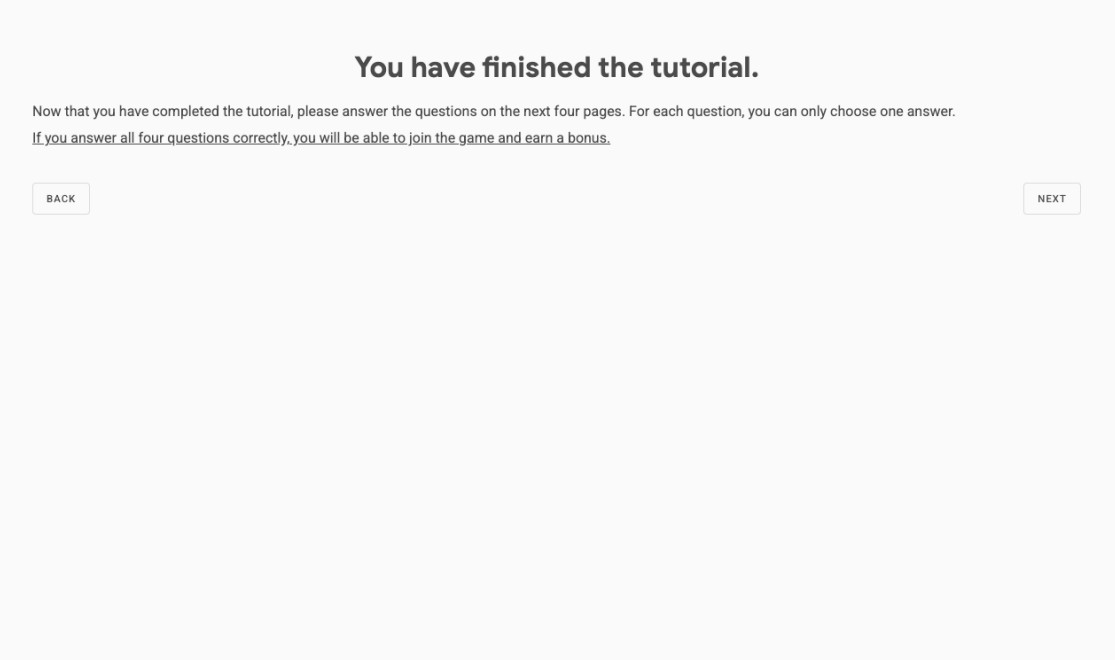

(b) Screen 10: Introduce comprehension test.

Figure 18: Screenshots of the participant interface for the "Deal or No Deal" study. (a) The participant reads tutorial information about the game. (b) The participant reads information about the comprehension test.

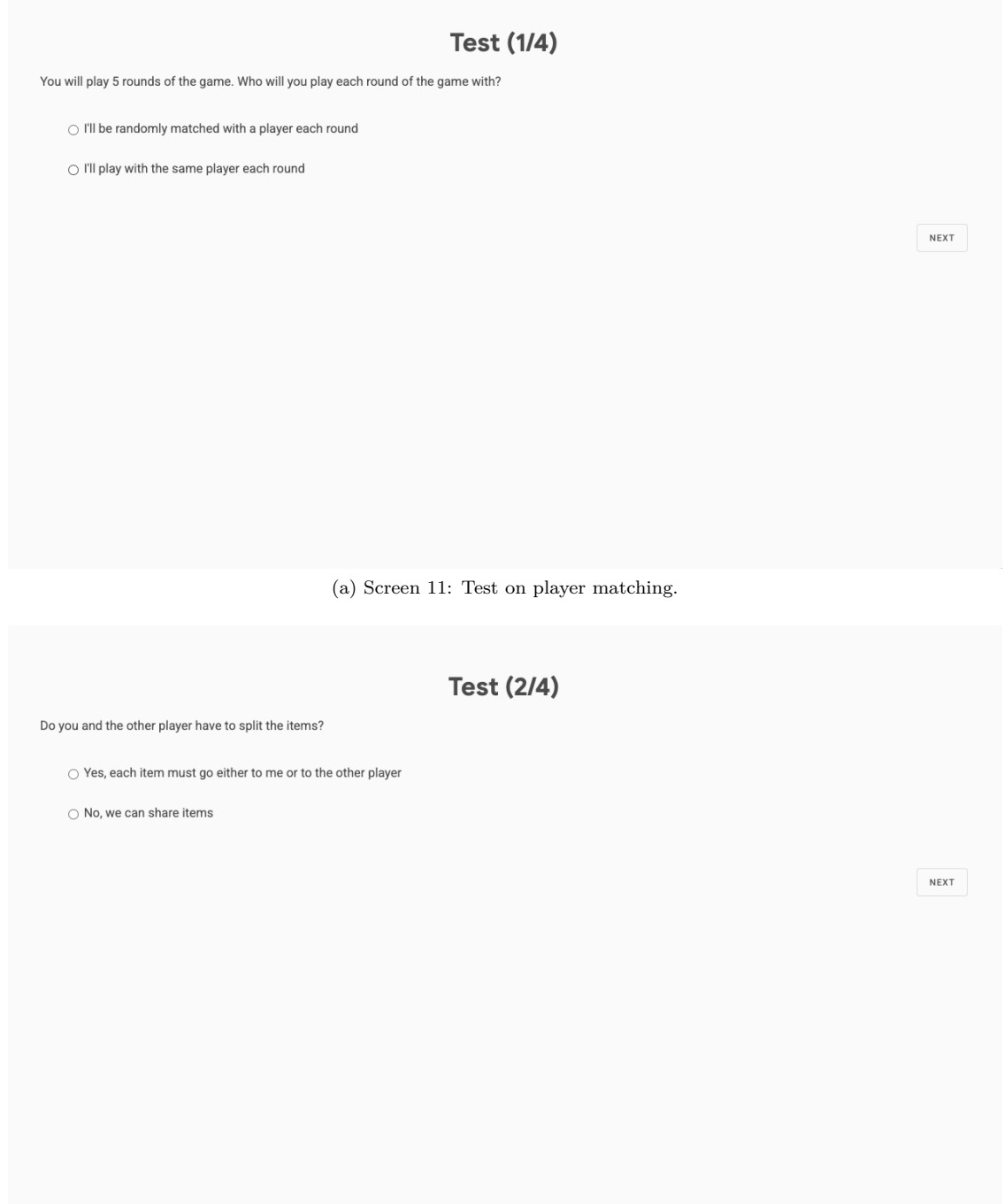

(a) Screen 11: Test on player matching.

(b) Screen 12: Test on item distribution.

Figure 19: Screenshots of the participant interface for the "Deal or No Deal" study. (a) The participant takes the comprehension test. (b) The participant takes the comprehension test.

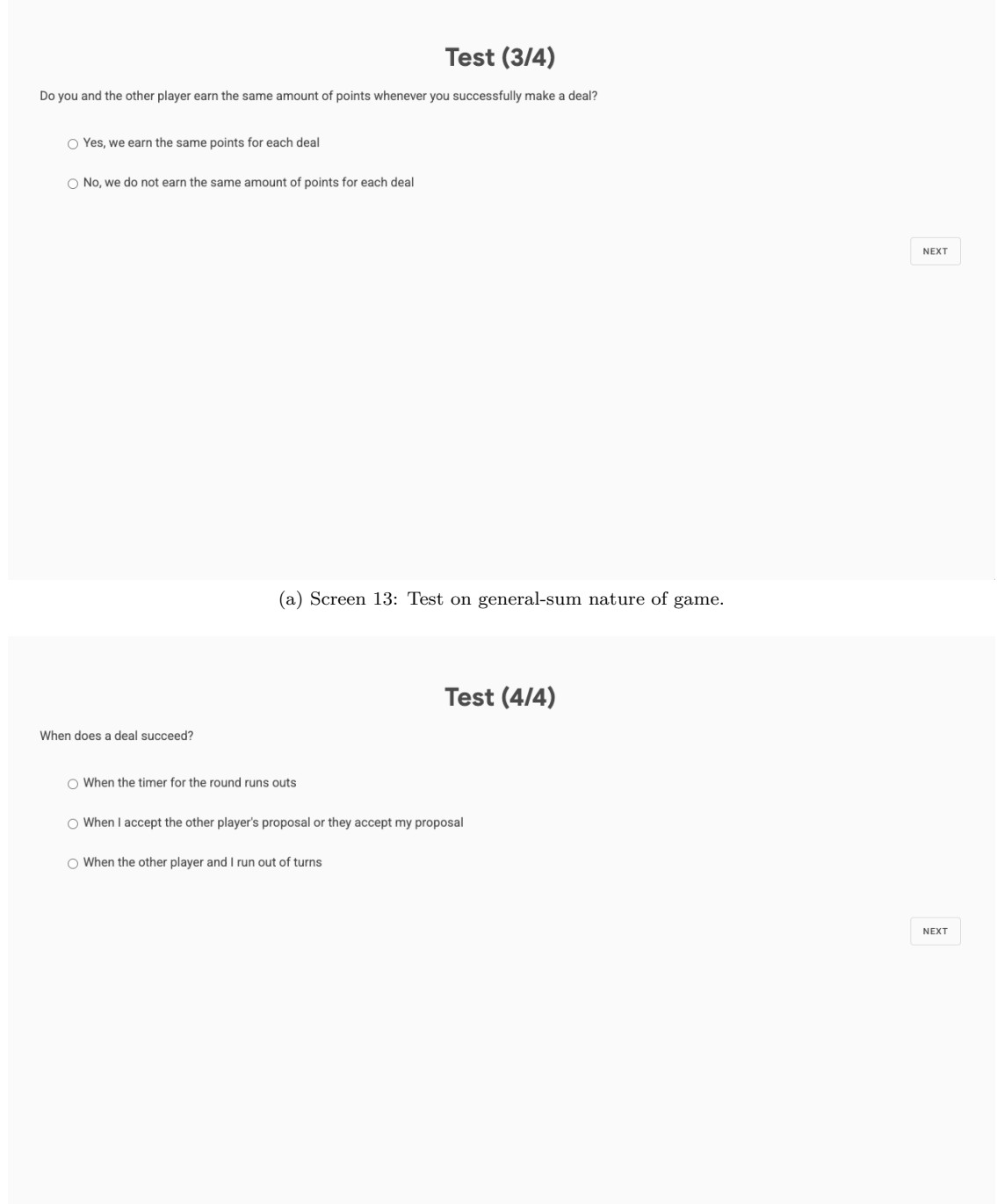

(a) Screen 13: Test on general-sum nature of game.

(b) Screen 14: Test on conditions for deal success.

Figure 20: Screenshots of the participant interface for the "Deal or No Deal" study. (a) The participant takes the comprehension test. (b) The participant takes the comprehension test.

**You passed the comprehension test!**

The game will start in 03:18.

Please be ready to play when this timer reaches zero. If you see an error message when the game starts, try refreshing the page.

Figure 21: Screenshots of the participant interface for the "Deal or No Deal" study. The participant sees their results for the comprehension test. If they answered all four questions correctly, the participant waits to be randomly assigned to a game session.

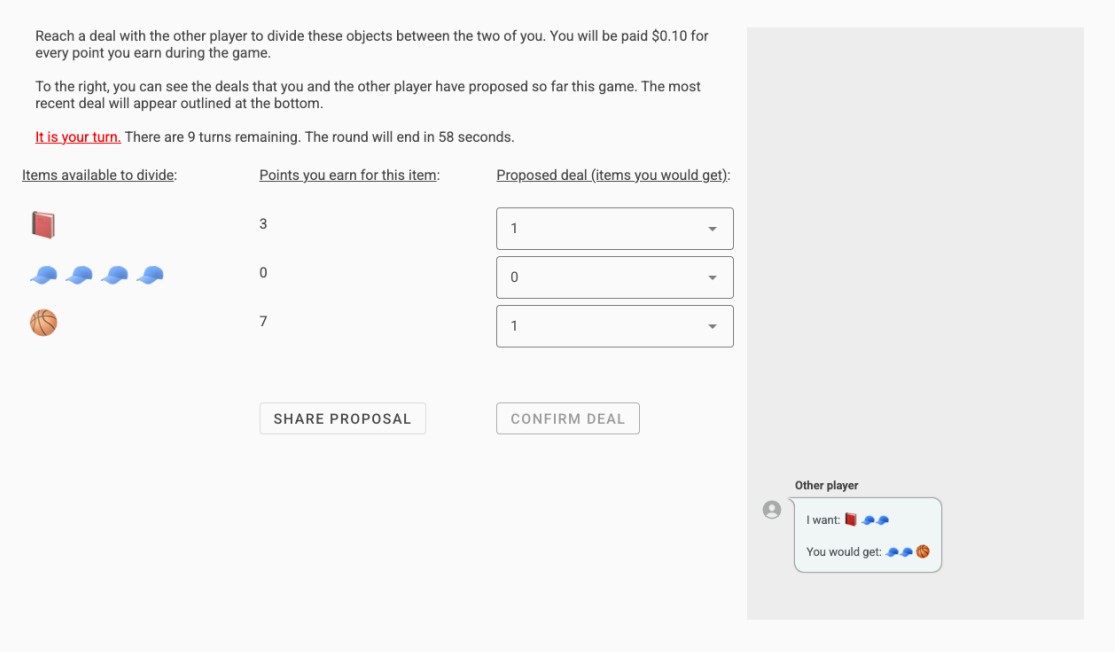

(a) Game screen: Prompt participant to take their turn.

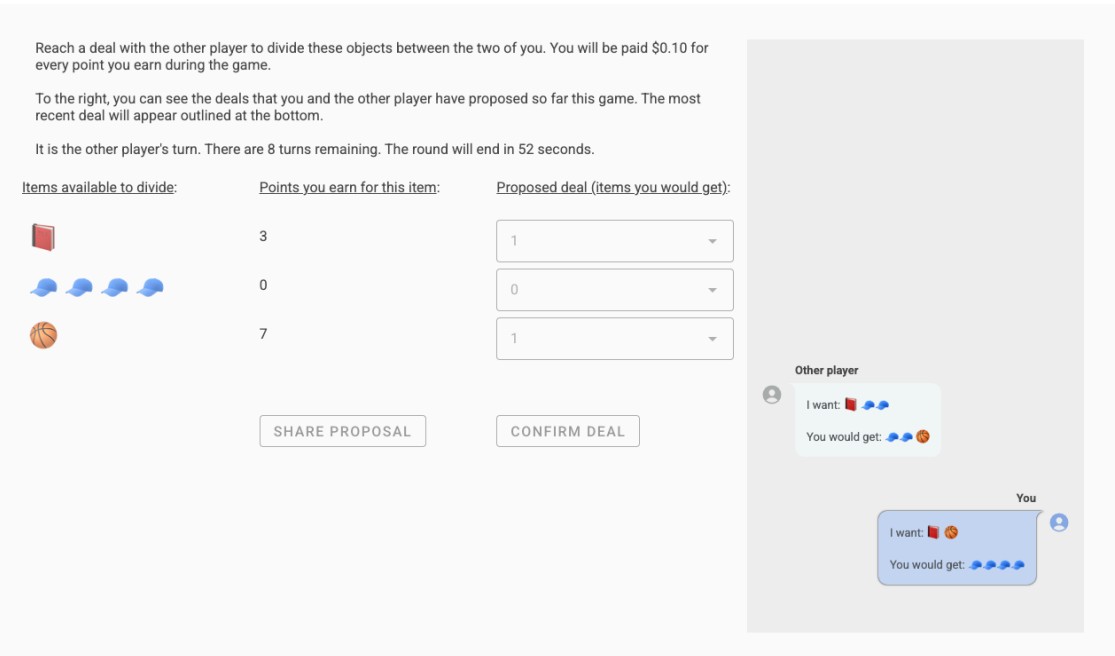

(b) Game screen: Wait for other player to take their turn.

Figure 22: Screenshots of the participant interface for the "Deal or No Deal" study. (a) The participant proposes a deal to the other player. (b) The other player chooses to confirm the deal or proposes a new deal.

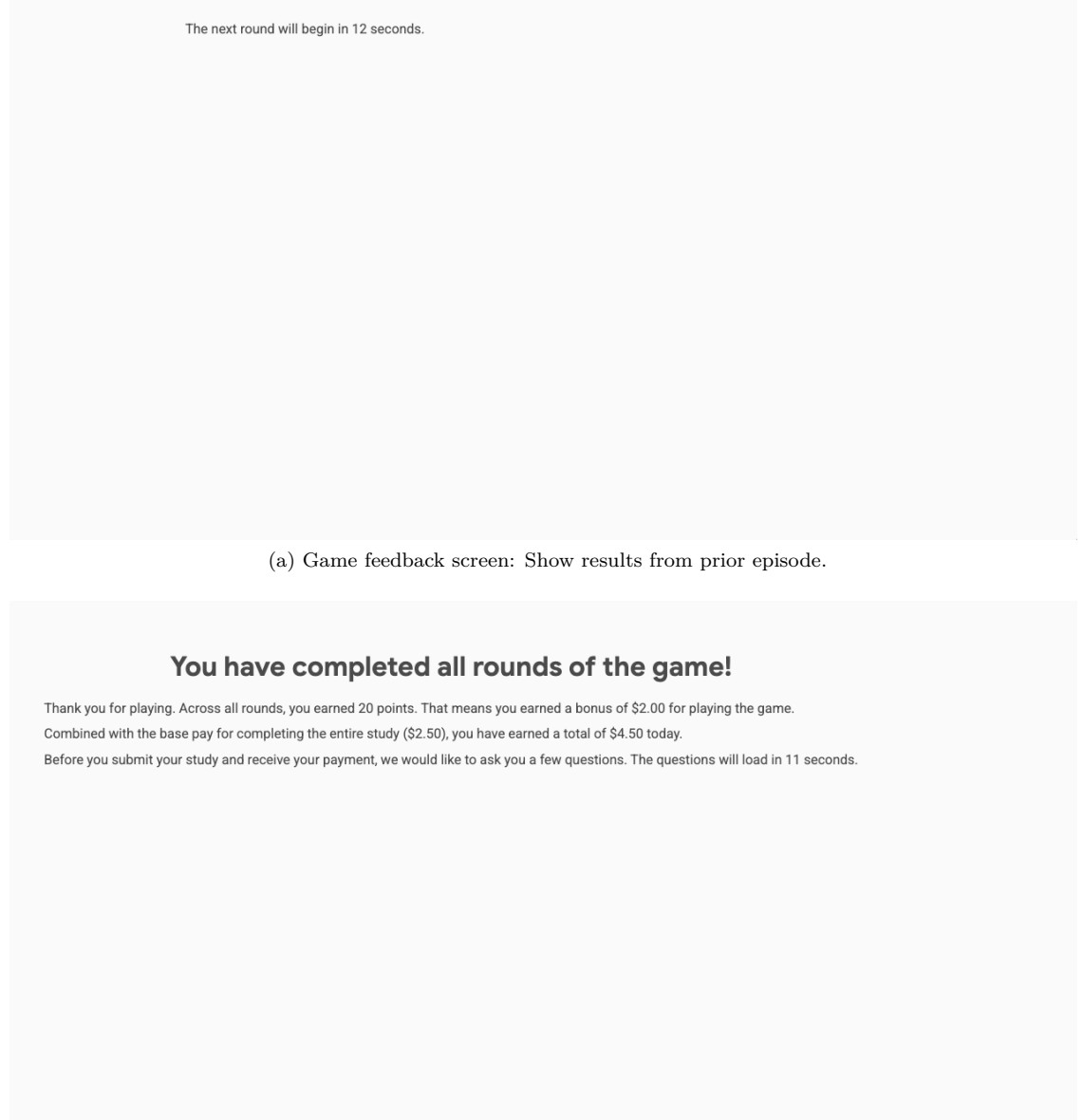

(a) Game feedback screen: Show results from prior episode.

(b) Study feedback screen: Show bonus and introduce post-task questionnaire.

Figure 23: Screenshots of the participant interface for the "Deal or No Deal" study. (a) The participant waits for the next game to start. (b) The participant sees their bonus and waits to begin the post-task questionnaire.

