# OpenReview forum: "Combining Tree-Search, Generative Models, and Nash Bargaining Concepts in Game-Theoretic Reinforcement Learning"
_TMLR — Rejected by TMLR_

### Review · Reviewer_6Whi · 2023-08-04

**Summary Of Contributions:**

This manuscript described a new system for solving general-sum, imperfect information game. The main technical novelty lies in the integration of tree-search and generative modeling into PSRO framework, and several new meta-strategy solvers. Experiments on several benchmark, including two negotiation games and one with human participants, demonstrate the effectiveness of the proposed method.

**Audience:**

Yes

**Broader Impact Concerns:**

The authors may add some discussion on the potential application of the proposed system and how it can affect the society.

**Claims And Evidence:**

No

**Requested Changes:**

* I suggest separating the Nash Bargain Solution concept into an individual paper, putting more theoretical analysis and empirical justification on the effectiveness of it. Theorem 4.1 is just a standard online convex optimization guarantee and I don’t find any interesting conclusion here.
* I would like to see more ablation studies on the effects of each individual components. Maybe it is already in the appendix but I believe the authors should make some proper reorganization to make the purpose of this paper more clear.

**Strengths And Weaknesses:**

### Strengths
Imperfect-information general-sum games are extremely hard to deal with, both in theory and in practice. This manuscript brings some new insight into this direction, with detailed experimental justification.

### Weaknesses
The proposed system has so many components and it is indeed hard to parse the contribution of each part. Specifically
* I believe that the MCTS + generative modeling part is independent of the concept of Nash Bargaining Solution, and I don’t quite get the motivation to put them in a single paper. I also would like to mention that I don’t see so many clear differences between the newly proposed meta strategy solvers and the existing one.
* For me I would like to see clear ablation studies on the effect of each components (MCTS, generative modeling). Currently the authors only make very brief discussion in Sec 5.3.1. As the main target of this paper is incorporating these components into PSRO, I believe such ablation studies are necessary.

---

> ### Author Response · Authors · 2023-10-02
> **Thanks for your review**
>
> We sincerely thank the reviewer for taking time to read our work. We now address your points as follows.
>
>
> $\bullet$ About the Nash Bargaining Solution section and the overall structure of our paper. We consider our contributions of NBS in Section 4 as well as our development of the new search algorithm in Section 3 are coherently unified under the PSRO-style game-theoretic RL theme. Both are important advancements for different components of PSRO. It is true that both the new search method and NBS results could be regarded as independent research results of their own. However, given our substantial developments on each different component of PSRO and the success of combining them together, we consider all our contributions to be coherently unified under the PSRO-styled game-theoretic RL theme, and TMLR as a journal is an appropriate venue for submitting our work.
>
>
> $\bullet$ Regarding the ablation studies. We sincerely apologize that we are unable to conduct more rigorous, thorough ablation studies in the near future than what were there in the paper, due to the current limitation of our resources for conducting more human experiments. However, we kindly remind the reviewer that our current results already more or less investigate the effects of different components. Section 5.3.1 studies the effect of search-based BR v.s. Non-searched-based BR, which shows the effectiveness of search. The effectiveness of different meta-strategy solvers are investigated in section 5.1 and 5.2, which shows MSSs like regret-matching or ADIDAS are better at reaching equilibria, while NBS and social-welfare are better at producing high social welfare profiles and reducing the Pareto gap. In our final human experiments, we selected four PSRO agents, where Comp1 and Comp2 did not employ search at test-time while Coop and Fair did. Although this may not be a perfect ablation study as the hyperparameter combinations are totally different, this may still suggest the effectiveness of search in the whole PSRO loop.
>
>
> Lastly, we agree adding a discussion section on society impact and AI safety is critical; we will add that in our paper.

---

### Review · Reviewer_PmXV · 2023-09-09

**Summary Of Contributions:**

This work presents a method for game-theoretic reinforcement learning. The core basis of this method is PSRO (Policy Space Response Oracle). This is a quite general framework, inspired by “double oracle” algorithms for normal form games. The general idea of PSRO is that given a pool of pure strategies (in this context, RL policies) in a game, one can compute an equilibrium mixed strategy distribution over these policies (the “meta-strategy”). Then, each player has some best response oracle that can calculate a new pure best response to these mixed strategies, which is added to the pool, after which the process repeats.

The contribution of this paper is to add new ideas to every aspect of the framework. The main contribution is in a fairly complicated best response oracle incorporating MCTS over information sets, and a generative model which approximates the probability distribution over full histories given the current infoset. Additionally the authors present several new meta-strategy solvers (e.g. coarse-correlated equilibria chosen to maximize the Nash bargaining score).

They test the components of these new contributions independently and in combination, and find that they achieve strong results on some small benchmark as well as much larger games. In particular, they test various alternative goals for the meta-strategy solver in very small games (e.g. Leduc poker); they test as well on a negotiation game called "Colored Trails" (in this case using DQN without their novel search techniques as the BR oracle). Finally on Deal or No Deal, a much larger scale game, they consider their full method and show that it achieves good performance faster than baselines (and sometimes to a level never reached by the baselines).

After this, the authors evaluate how their agents play against actual humans in “Deal or No Deal”.

**Audience:**

Yes

**Broader Impact Concerns:**

For work like this, I don't see any significant broader impact concerns.

**Claims And Evidence:**

Yes

**Requested Changes:**

I have no requested changes for the authors, although if they wish they can respond to any of the issues raised in "weaknesses".

**Strengths And Weaknesses:**

Strengths:

- There are a wide range of reasonably convincing experiments, including ablation studies of the different components of the algorithm.

- The details of the method are made clear and explicit.

Weaknesses:

- This is definitely an “A+B+C” paper, for the most part plugging together pieces that had existed in previous work.

- The method seems complex, with a large number of moving parts. The experiments do try to grapple with this, but there are still a lot of pieces.

- Their full algorithm with all contributed components is only tested on one game

- Presentation is quite unclear at times in some of the experiments, e.g. which baselines represent "instantiations" of the framework from this paper, and which are just other techniques entirely (is ADIDAS one of these?)

- I'm not an expert in these "play social dilemma experiments with humans" or their methodology, but the methodology described in that section for choosing agents seems a little ad-hoc.

---

> ### Author Response · Authors · 2023-10-02
> **Thanks for your review**
>
> We sincerely thank the reviewer for taking time to read our work. We now address your points as follows.
>
> $\bullet$ About the overall structure and content of our paper. We respectfully disagree with your point that our work is merely a combination of what had existed in previous literature. Instead, our work made several novel advancements in both the best response step and meta-strategy step in the PSRO framework which had not been developed in previous literature. For the best response component, we employ an imperfect information search as the best response step, where we incorporate a deep generative model to handle large belief spaces. This could be regarded as an independent research contribution of its own; we effectively developed a new powerful POMDP planning algorithm that supersedes the previous ones. Similarly for the meta-strategy step, we proposed solutions based on cooperative game theory, where previous works focused only on solutions based on non-cooperative game theory. The results in Section 4 also could be regarded as an independent research contribution of its own, where we present new efficient algorithms with theoretical analysis. We experimentally demonstrated the effectiveness of the new search algorithm in Section 5.3.1 and the new MSSs in Section 5.1 and 5.2, respectively, and present the power of combining them together in our human experiments in section 5.3. Given our substantial developments on each component of PSRO and the success of combining them together, we consider all our contributions to be coherently unified under the PSRO-styled game-theoretic RL theme, and TMLR as a journal is an appropriate venue for submitting our work.
>
>
> $\bullet$ About the experimental setups, and how we chose the agents. For the results of Section 5.1 and 5.2, we are comparing different “instaniations” of the whole methods, which represents different solution concepts in the whole PSRO loop computed by MSS; and ADIDAS is one of these. In section 5.3 we compared PSRO agents with self-play DQN agents. We respectfully disagree that our agent-selection process is ad-hoc. We employ empirical game theoretic analysis [1] and selected the agent based on individual scores, social welfare scores and inequity aversion. This process is purely generic and purely game theoretic without any domain knowledge of the game.
>
> Lastly, we acknowledge the current complex nature of our exposition. We sincerely solicit the reviewer’s opinions on where and how to improve it and we are open to any change to make our results clearer.
>
> [1] “Methods for empirical game theoretic analysis”, Wellman, AAAI’06

---

### Review · Reviewer_CkHW · 2023-09-20

**Summary Of Contributions:**

The paper focuses on the problem of solving general sum imperfect information games using reinforcement learning. To this end, the authors study and extend the combination of population based deep reinforcement learning approach with tree search procedure that has previously achieved significant success in purely adversarial, perfect information games.  Specifically, the authors identify two challenges in making this extension: existence of alternative equilibria in general sum games and presence of imperfect information that requires to build model of belief over world states. To address these challenges, the paper consider a population based training regime for multi-agent reinforcement learning and focuses on the previously proposed approach of Policy Response Space Oracles (PSRO) in this space. In its earlier form, PSRO comprises of two basic steps: Meta Strategy solver (MSS) step that computes a distribution over strategies and Best Response Step that computes the best response policies using reinforcement learning agains the strategies found in MSS step. This work studies and extends PSRO to make it applicable to general sum imperfect information games in the following ways: In addition to previously available meta strategy solvers, this paper proposes three more solvers inspired from the bargaining theory that focuses on optimizing for pareto efficiency, coarse correlated equilibrium and social welfare.  Next, they consider Approximate Best Response computation approach for the best response step. Specifically they augment the reinforcement learning procedure to compute best response step with the information Set Monte Carlo tree Search that works on improving the trajectories that are used to train the RL procedure. Further, to account for imperfect information, they also employ a generative model that represent the belief over the world states  during search. Experiments are done on few benchmark games and in two negotiation  games where a comparison is also made with human performance on one of these two games. These experiments are proposed to study the effect of various choices of meta-strategy solvers and generative model. In particular, they assess the capacity of PSRO with different MSS to serve as Nash equilibrium solvers and the performance of trained agents in negotiation game under different metrics.

**Audience:**

Yes

**Broader Impact Concerns:**

General equilibrium finding algorithms that can be deployed on large scale multi-agent systems have numerous applications including but not limited to negotiations, security, auctions, etc. Given the proposed work particularly focuses on supporting imperfect information and large state spaces, such approaches, while being very useful, carry potential risk for being used for cheating and gaining undue advantage in such systems. Hence, I believe that a broader impact statement and an exploratory discussion of such approaches from the perspective of AI safety is a critical piece for publishing such works.

**Claims And Evidence:**

Yes

**Requested Changes:**

- I find the overall combination to be less interesting contribution except for its implication. If I understand correctly (and as described briefly in the paper), the key impact of this approach is that the generative model enables diversity in the opponent strategies where sampling different representations results in simulating a different type of opponent. This diversity induction is a subtle but strong point, however this has not been analyzed in the paper rigorously. Could the authors elaborate  on this point and provide analytical or empirical studies of such diverse strategies learned by the model? For instance, it would be interesting to see what types of opponent strategies are sampled in the Deal-or-No-Deal game.
- The authors introduce an expensive search step as a part of PSRO. It is shown that this method outperforms other approaches that do not use this search based method but then it becomes imperative to compare the efficiency of this approach. Does the proposed approach take too much time in search to come up with reasonable solution compared to non-search based approaches? Or another way to look at it is what kind of solutions this approach ends up with in time that non-search based approaches require to converge?
- How scalable is the approach in terms of number of players?
- Can this approach be considered as. plug-and-play where one can easily replace Nash Bargaining solution with any other solution concept? Is there a class of solution concept or conditions that need to be met so as to apply this approach?
- What are some of the immediate limitations of this approach?
- I also find the overall presentation of the paper to be cluttered and difficult to follow in some parts. It would help to use more diagrams to explain existing concepts and clearly mark out new contributions plus use table for notations.

**Strengths And Weaknesses:**

### Strengths
---

- The idea of replacing purely RL based approach in PSRO to compute best response with the Approximate Best Response approach (RL + IS-MCTS) to support general sum games is an interesting combination.  To this end, the paper provides a very good example of organically combining existing techniques to address challenges in previously unsolved problems.
- The use of deep generative model to further enhance ABR so as to learn representations of belief states is a useful feature to address intractabilities that come with large games.
- The adoption of Nash bargaining as a Meta Strategy solver provides a good recipe to consider different game-theoretic concepts as part of this population based training framework for supporting different game settings.
- The experiments on Colored Trails and Deal or no Deal domains provide useful insights into the effectiveness of this approach.
- The proposed approach seems to effectively negotiate with humans even when human participation is not considered during training which is a good result.

### Weaknesses
---

- Overall, I am a bit concerned about the contributions of this paper. The paper is focussed on solving imperfect information general sum games. PSRO has been previously shown to work for general sum games and ABR based design provides strength towards improving the strategies which is also known. Given this, the key technical contribution is the use of deep generative model to represent belief states (estimate the posterior) instead of computing exact posterior.
- However, the treatment of deep generative model is very hand way in the paper. For instance, it is not clear what kind of representations are learned by such a model, how far they are from the exact posterior and how does it help the downstream search of the strategies. Experiments studying these properties in detail would be useful. Further, it is not clear if the approach is scalable to more than 2-3 player games. If I understand correctly, as the opponents are fixed, at episode, the approach behaves as a single agent learning. However, the size of belief space would tend to increase with the increase in total number of players. And this may lead to difficulties with training the generative model or making it efficient, which is the key purpose of introducing it.
- In terms of meta-strategy solvers, it is good to see the recipe of how to use a new concept from game theory to inform the learned strategies, however, the paper would benefit by considering more examples of different situations (e.g.  reciprocity) and analyze whether this kind of approach generalizes across different games and settings.
- Specific to Nash bargaining solution, it appears (at least in the Colored Trails experiment) that NBS based solver may or may not better than maximizing social welfare. While this is a purely empirical question in my opinion, this needs to be studied in more detail as use of NBS is described as one of the contributions of this work.

---

> ### Author Response · Authors · 2023-10-02
> **Thanks for your review**
>
> We sincerely thank the reviewer for taking time to read our work. We now address your points as follows. As some of your questions are related to each other we may bundle the answers to them in one section.
>
>
> $\bullet$ The generality of the search method. We consider the idea of incorporating a deep generative model learning loop is very general for search in imperfect information games. The design of the neural architecture, however, does require inductive biases and domain knowledge of the game. For example in the DoND game we tested on, the deep generative model is designed to predict the opponent’s private valuations, which are the only hidden information of this game. Since the valuations are integer vectors, we designed the output to be logits that define distributions over the integer values. And we train the neural nets via a cross-entropy classification loss. To facilitate the learning process, we consider it better and unavoidable to have a handy design for different games.
>
>
> $\bullet$ The scalability of the search method. We consider the bottleneck does not necessarily depend on the number of players, but on the hidden information of the game that we need to infer. Another way to view it is that when we fix other-players’ strategies, it becomes a planning problem in POMDP where the hidden state incorporates anything that is not observed by the search agent. Therefore again the design of the neural representations is very crucial and using domain knowledge will usually help a lot to facilitate the learning process. For example, in the DoND the only essential things we need to predict are opponent’s valuation vectors, so it is redundant to incorporate other state information such as offers and item pool as part of the prediction as they have already been included in the infostate input. And we consider it will facilitate the learning process. Therefore similarly for games with more players, it does require the programmers to identify an efficient representation of the hidden information, and that representation does not have to be harmed by a growing number of players. For example, in multiplayer poker if there are certain correlations among different players’ hands, then multiple neural heads with certain parameter sharing may help to learn better representations. We consider leveraging such domain knowledge and the expressiveness of deep neural nets is the best way we can come up with to represent belief states in imperfect information games [1, 2, 3], as previous approaches like particle filtering suffer from the same issue without a clear way out.
>
>
> $\bullet$ Meta-strategy solvers, Nash Bargaining Solutions and solution concepts. In our work we considered a variety of solution concepts computed by different MSSs, and we evaluated all of them in our experiments. Section 5.1 demonstrates the performances of all 16 different MSSs on a set of benchmark games; Section 5.2 found out among all these 16 MSSs the NBS-based and social-welfare based are the best at reducing the Pareto gap; and in Section 5.3 we trained a variety of agents with different MSSs and selected four of the best ones to submit to the human experiments. Our framework is general enough to plug-in any kinds of solution concepts, which we empirically did with Nash equilibrium, correlated equilibrium and NBS-based. Regarding the performances of NBS MSS, from the experiments it is true that it is not always better than social welfare in terms of reducing the Pareto gap, but it still usually achieves quite high performances among all 16 MSSs. We made several similar observations in Section 5.1 as well. Notice that there are very few previous works that studies NBS notions in non-cooperative normal-form games; the only work we are aware of is [4], which proposes a quadratic program for solving only 2-player NBS. Our development in Section 4 could be an independent research contribution of its own, where we propose a more efficient projected gradient ascent algorithm for N>2 cases, and analyze the max-NBS-(C)CE solution concepts. The convergence guarantee of PSRO does not change, i.e., in the long run the algorithm will converge to the MSS solution.
>
>
> Lastly, we acknowledge the current complex nature of our exposition. We sincerely solicit the reviewer’s opinions on where and how to improve it and we are open to any change to make our results clearer. We agree adding an impact statement and discussion of AI safety is critical; we will add that in our paper.
>
>
>
>
> [1] “Deep Counterfactual Regret Minimization”, Brown et. al. ICML’2018
>
> [2] “Deepstack: Expert-level artificial intelligence in heads-up no-limit poker”, Moravčík el. al.  Nature
>
> [3] “Player of Games” Schmid et. al. 2021
>
> [4] Quadratic programs and general-sum games. In Game Theory: Penn State Math 486 Lecture Notes, Christopher Griffin.

---

> > ### Comment · Reviewer_CkHW · 2023-11-22
> > **Thank you for your response**
> >
> > I thank the authors for providing detailed responses to my concerns and questions. I highly appreciate the addition of the broader impact statement and willingness to improve on the complex nature of the exposition. To this point, a couple of recommendations that may be helpful: (i) I wonder if it would be a good idea to abstract away long textual descriptions with figures or other visual elements representing the overall approach, especially the components that belong to previous work. Further, it might help to use tables or separate boxes to list out symbol usage and formula like PUCT. (ii) It takes a few reads to figure out what are the exact contributions of the manuscript which makes it less interesting to read.  Some form of color coded text and/or separately highlighted contributions would go a long way in distilling the exact contributions of the paper compared to previous works. I am happy with the response and clarifications regarding 16 different MSS provided by the reviewer. I am still concerned about the generality and scalability of the approach and at this point, the treatment of deep generative model in the paper is not convincing as an important contribution. I think I understand this work fairly well and I do not have any further questions for the authors. Thank you once again for the research contribution and spending time and efforts in responding to my review comments. I hope this discussion will contribute towards the improvement of future versions of this work.

---

### Decision · Action_Editor_mD4c · 2023-11-30

**Recommendation:** Reject

**Comment:**

This paper incorporates several new ingredients to the PSRO framework: a Monte-Carlo Tree Search procedure, a deep generative model for handling imperfect information, as well as a Nash-Bargaining solution as the meta-strategy solver.

The opinions about this paper are mixed among reviewers. While the reviewers generally agree that the proposed approach makes progress and some of the new ingredients (such as the deep generative model) are natural, there are serious concerns about (1) lack of understanding on the significance of each individual component; (2) the presentation and paper organization being cluttered and some of the experimental results being difficult to comprehend; (3) some concerns with the experimental setup in Table 3, which evaluates 4 selected agents out of the learned population, and also with the target (objective) of improvement being slightly unclear.

In general, I think the *style* of this paper is a good fit for TMLR: incorporating new techniques into a widely used algorithm (PSRO) to achieve improved performance. In my opinion, an ideal goal for such a kind of paper would be adoption in future research. However, at the current stage of this paper, the weaknesses in the results and presentations mentioned above may significantly limit the impact and adoption of the proposed approach. And this can be exacerbated by the fact the proposed approach contains many moving pieces.

I thank the authors for submitting this work to TMLR and for engaging in the discussions. I encourage the authors to carefully polish the presentation and consider adding some more experiments to strengthen the claims (e.g. significance of each component of the algorithm) according to the reviewers' suggestions, which would make it a much stronger submission to TMLR or another top venue.

**Audience:**

This paper makes contributions in imperfect-information games and PSRO algorithms, both of which are of interest to the games community.

**Claims And Evidence:**

There are concerns about the lack of understanding on the significance of each individual component, many of the presentation being cluttered and difficult to comprehend, and some concerns with the experimental setup. (See more details in the comment section.)

**Resubmission Of Major Revision:**

The authors may consider submitting a major revision at a later time.